# VaMP: Variational Multi-Modal Prompt Learning for Vision-Language Models

**Silin Cheng**     **Kai Han**[*]
Visual AI Lab, The University of Hong Kong
hnslcheng@connect.hku.hk     kaihanx@hku.hk

## Abstract

Vision-language models (VLMs), such as CLIP, have shown strong generalization under zero-shot settings, yet adapting them to downstream tasks with limited supervision remains a significant challenge. Existing multi-modal prompt learning methods typically rely on fixed, shared prompts and deterministic parameters, which limits their ability to capture instance-level variation or model uncertainty across diverse tasks and domains. To tackle this issue, we propose a novel **Va**riational **M**ulti-Modal **P**rompt Learning (VaMP) framework that enables sample-specific, uncertainty-aware prompt tuning in multi-modal representation learning. VaMP generates instance-conditioned prompts by sampling from a learned posterior distribution, allowing the model to personalize its behavior based on input content. To further enhance the integration of local and global semantics, we introduce a class-aware prior derived from the instance representation and class prototype. Building upon these, we formulate prompt tuning as variational inference over latent prompt representations and train the entire framework end-to-end through reparameterized sampling. Experiments on few-shot and domain generalization benchmarks show that VaMP achieves state-of-the-art performance, highlighting the benefits of modeling both uncertainty and task structure in our method. Project page: https://visual-ai.github.io/vamp

## 1 Introduction

Vision-Language Models (VLMs), such as CLIP [1], have achieved impressive performance across a wide range of visual recognition tasks through multi-modal representation learning. Their ability to align images and texts in a shared embedding space enables strong zero-shot transfer. However, their large-scale parameters and the scarcity of training data, particularly in few-shot settings, make full model fine-tuning computationally expensive and prone to overfitting on downstream tasks.

To address this, prompt learning has emerged as a parameter-efficient alternative, where a small number of learnable tokens are prepended to the input to steer the frozen model toward task-specific behavior [2, 3, 4, 5, 6, 7, 8]. While effective, existing multi-modal prompt tuning methods typically rely on fixed, shared prompts that are applied uniformly across all samples. Such methods are inherently deterministic and lack the flexibility to adapt to instance-level variations or model uncertainty, limiting their generalization to unseen tasks and domains [9, 10].

While recent work has explored uncertainty-aware prompt tuning, most existing approaches remain limited in scope. Bayesian Prompt Learning introduces uncertainty modeling in text-only prompts, and Any-Shift Prompting leverages variational inference to enhance robustness across distribution shifts. However, these methods suffer from key limitations. First, by restricting variational modeling to input-level prompts with global latent variables, they fail to capture hierarchical feature interactions

---

[*]Corresponding Author

39th Conference on Neural Information Processing Systems (NeurIPS 2025).

or fine-grained, token-level semantic variations. Second, their text-only prompt tuning overlooks valuable visual information that could enhance cross-modal alignment. Finally, the standard Gaussian prior, shared across all classes, fails to capture inter-class variations, resulting in less discriminative prompt distributions. Consequently, these models are inadequate for capturing fine-grained, input-specific variations, particularly in vision-language tasks where precise alignment is critical.

To overcome these limitations, we introduce a novel **Va**riational **M**ulti-Modal **P**rompt Learning (VaMP) framework that enables sample-specific, uncertainty-aware prompt tuning for vision-language models. We make three key contributions: First, we introduce token-wise variational modeling across multiple intermediate network layers. This approach treats individual prompt tokens as latent variables, enabling the model to capture fine-grained semantic relationships at multiple abstraction levels and improve generalization in low-data and out-of-distribution scenarios. Second, our multi-modal design incorporates both visual and textual signals when inferring posterior distributions, creating more aligned cross-modal representations. Third, by employing class-aware priors instead of standard Gaussian distributions, VaMP generates more discriminative prompts that better capture category-specific features and decision boundaries.

We evaluate VaMP on three challenging adaptation settings: base-to-new generalization, domain generalization and cross-dataset generalization. Our method consistently outperforms strong multi-modal prompt baselines while maintaining high parameter efficiency. Ablation studies further confirm the effectiveness of each component, including the variational modeling and task-aware prior.

## 2   Related Work

**Pre-trained Vision-Language Models.** Pre-trained vision-language models (VLMs) [1, 11, 12, 13] have gained significant attention for their strong performance across diverse vision-language tasks. These models typically follow four training paradigms: 1) masked language modeling [14, 15]; 2) masked region prediction [16, 17]; 3) image-text matching [16, 14]; and 4) contrastive learning [1, 11, 18, 19]. While VLMs provide robust, generalized representations, adapting them to downstream tasks remains challenging. Recent studies show that tailored approaches significantly improve performance in specific domains, such as few-shot image recognition [20, 21], object detection [22, 23, 24, 25, 26, 27], semantic segmentation [28, 29, 30, 31, 32] and visual grounding [33, 34]. In this work, we focus on adapting vision-language models for few-shot and zero-shot visual recognition tasks.

**Prompt Tuning.** Instructions provided to language models as text prompts enable task-specific understanding and performance in VLMs. These prompts, either manually designed or automatically optimized through "Prompt Learning" (originally from NLP [35, 36, 37]), have been adapted for computer vision in three primary forms: textual prompt leanring [2, 3, 38, 39, 40, 41, 42, 43, 44, 45, 46] that fine-tune CLIP's [1] by optimizing continuous prompt vectors in its language branch; visual prompt learning [4, 47, 48, 49, 50] that optimize task-specific learnable inputs in the visual input space while keeping pre-trained backbones frozen; and multi-modal prompt learning [5, 51, 52, 53, 54, 55, 6] that enhance alignment by applying prompts to both vision and language branches. Our work advances this research by introducing a variational framework for multi-modal prompt tuning, enabling sample-specific, uncertainty-aware prompt tuning with structured guidance from both visual inputs and class-level semantics.

**Variational Inference.** Variational inference has been widely applied to computer vision tasks, such as image generation [56, 57, 58, 59], action recognition [60], instance segmentation [61], anomaly detection [62], depth estimation [63], few-shot learning [64, 65, 66, 67], and domain generalization [68, 69]. Recently, variational inference has been applied to prompt learning to mitigate overfitting in low-shot settings and improve generalization. For example, Bayesian Prompt Learning [9] captures uncertainty by sampling prompts from learned distributions, while Any-Shift Prompting [10] uses a hierarchical probabilistic framework to model distribution shifts and generate adaptive prompts without test-time optimization. However, both methods are limited to the text modality and rely on globally shared prompts, which restrict their ability to capture fine-grained, input-specific variations. To overcome these limitations, we propose a probabilistic framework that combines multi-modal prompt tuning with variational modeling and sample-specific adaptation.

## 3  Preliminary

### 3.1  Revisiting CLIP

Our work builds upon the pre-trained vision-language model, CLIP [1], which comprises both a text encoder and a vision encoder. Following previous prompt-learning methods [2, 3, 5, 8], we adopt a ViT-based CLIP model that encodes both images $I \in \mathbb{R}^{H \times W \times 3}$ and text descriptions.

**Encoding Image.** The image encoder $V$ consists of $K$ transformer layers, denoted as $\{V_i\}_{i=0}^{K-1}$. It first divides the input image $I$ into $B$ non-overlapping patches. These patches are then projected into embeddings $e_0 \in \mathbb{R}^{B \times d_v}$, where $d_v$ represents the embedding dimension. Subsequently, patch embeddings, along with a class token $c_i$, are sequentially processed through the transformer blocks:

$$[c_{i+1}, e_{i+1}] = V_i([c_i, e_i]) \quad i = 0, 1, \cdots, K - 1. \tag{1}$$

The final image representation $x$ is obtained by applying a linear projection to the last layer's class token, mapping it into the shared vision-language embedding space:

$$f_x = F_{\text{img}}(c_K) \quad f_x \in \mathbb{R}^{d_{vl}}. \tag{2}$$

**Encoding Text.** The text encoder converts tokenized words into embeddings $w_0 = [w_0^1, w_0^2, \cdots, w_0^N] \in \mathbb{R}^{N \times d_l}$, which are processed through $K$ transformer layers:

$$[w_{i+1}] = L_i(w_i) \quad i = 0, 1, \cdots, K - 1. \tag{3}$$

Similarly, the text representation $t$ is obtained through a linear projection of the final embedding of the last token:

$$t = F_{\text{txt}}(w_K^N) \quad t \in \mathbb{R}^{d_{vl}}. \tag{4}$$

**Zero-shot Classification.** For classification, hand-crafted text prompts (*e.g.*, "a photo of a <category>") with class labels $y \in \{1, 2, \ldots C\}$ are used. The prediction $\hat{y}$ for image $I$ is determined by the highest cosine similarity:

$$\hat{y} = \arg\max_y \frac{\exp(\text{sim}(f_x, t_y)/\tau)}{\sum_{i=1}^C \exp(\text{sim}(f_x, t_i)/\tau)}, \tag{5}$$

where $\tau$ is the temperature coefficient.

### 3.2  Multi-Modal Prompt Learning

Multi-modal prompt learning methods [5, 70, 8] extend text-only prompt learning approaches [2, 3] by jointly tuning the text and image prompts to achieve improved alignment for downstream tasks. These methods typically specify $H$ consecutive transformer layers, beginning from the $J$-th layer, for prompt tuning, while leaving the remaining transformer layers fixed (where $J \geq 0$ and $H < K - J + 1$). For example, MaPLe [5] focuses on prompt tuning the shallow layers ($J = 0, H = 9$), whereas MMRL [8] applies prompt tuning to deeper layers ($J = 5, H = 7$).

**Deep Language Prompting.** In the text branch, we introduce learnable prompts $z_i \in \mathbb{R}^{M \times d_l}$, consisting of $M$ tokens, into the $i$-th transformer layer for prompt tuning. For each layer $i$ from $J$ to $J + H - 1$, these tokens are concatenated with the original token embeddings and fed into the $i$-th transformer layer to generate inputs for the next layer:

$$[\_, w_{i+1}] = L_{i+1}([z_i, w_i]), \tag{6}$$

while layers outside this range (from 0 to $J - 1$ and from $J + H$ to $K - 1$) remain unchanged:

$$[w_{i+1}] = L_{i+1}([w_i]). \tag{7}$$

Finally, the text representation is derived via Eq. 4.

$$[\_, w_{i+1}] = L_{i+1}([z_i, w_i]) \tag{8}$$

Meanwhile, layers outside this range (from 0 to $J - 1$ and from $J + H$ to $K - 1$) remain unchanged:

$$[w_{i+1}] = L_{i+1}([w_i]). \tag{9}$$

The final text representation is obtained through Eq. 4.

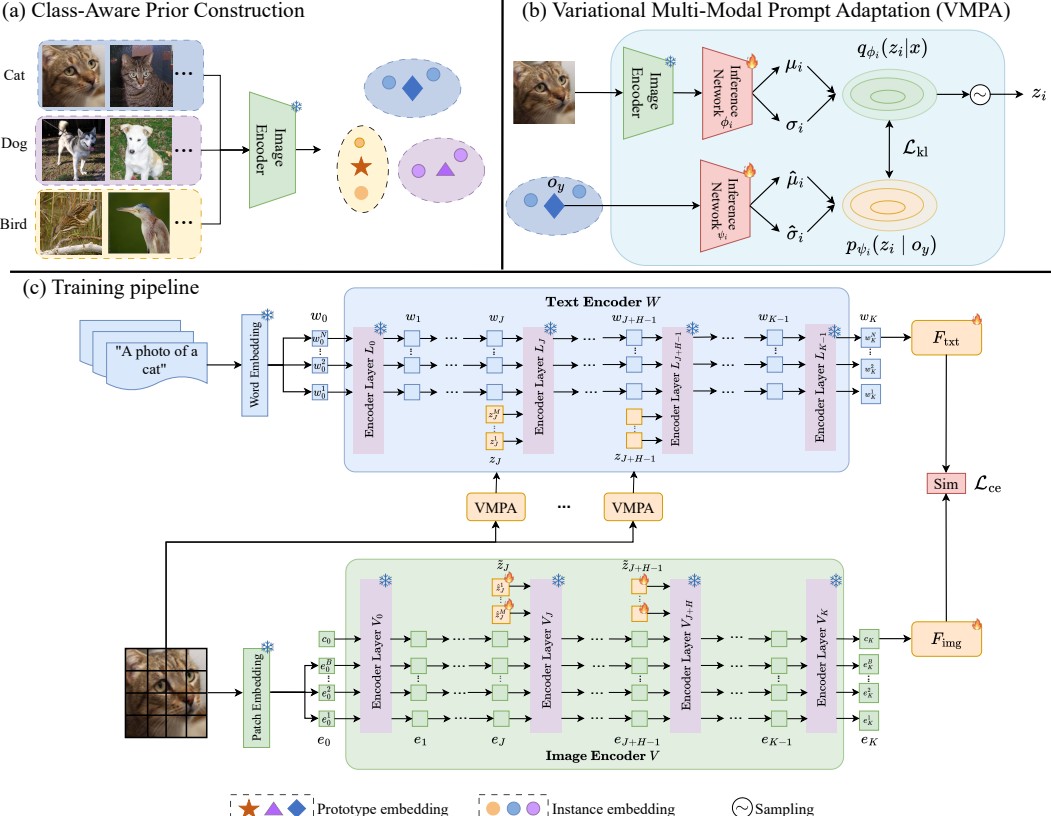

**Figure 1: Overview of the VaMP framework.** **(a) Class-Aware Prior Construction:** Utilizing CLIP's frozen image encoder to process training samples, generating offline class prototypes for subsequent adaptation. **(b) Variational Multi-Modal Prompt Adaptation (VMPA):** Variational modeling mechanism where image-conditioned posterior $q_\phi(z_i \mid x)$ and class prototype-based prior $p_\psi(z_i \mid c_y)$ are aligned through KL divergence regularization of latent prompt distributions. **(c) Training Pipeline:** Full architecture of our proposed VaMP framework.

**Deep Vision Prompting.** In the visual branch, M learnable tokens $\tilde{z}_i \in \mathbb{R}^{M \times d_v}$ are inserted into the $i$-th transformer layer for prompt tuning. The exact computation of these visual tokens depends on the specific multi-modal prompt learning method being used. For instance, in MaPLe [5], these visual tokens are generated from the language prompts via a linear transformation implemented with an MLP. In contrast, MMRL [8] obtains visual tokens from a shared latent space—a set of learnable tokens initialized by sampling from a Gaussian distribution—and then uses separate linear projection functions (also implemented with MLPs) to generate modality-specific prompts.

For each layer $i$ from $J$ to $J + H - 1$, these generated tokens are concatenated with the original patch token embeddings and fed into the $i$-th transformer layer to produce inputs for the next layer.

$$[c_{i+1}, e_{i+1}, \_] = V_{i+1}([c_i, e_i, \tilde{z}_i]).\tag{10}$$

For the remaining layers, the original operation without prompts is preserved:

$$[c_{i+1}, e_{i+1}] = V_{i+1}([c_i, e_i]).\tag{11}$$

The final image representation follows the same projection process outlined in Eq. 2. During inference, predictions follow standard classification procedures based on similarity.

## 4 Method

We propose VaMP—a variational multi-modal prompt learning framework that enables sample-specific, uncertainty-aware, and structured tuning within vision-language models. Our method

consists of three key components: (i) *sample-specific multi-modal prompt generation*, where image-conditioned prompts are injected across multiple transformer layers; (ii) *variational prompt adaptation for multi-modal representation learning*, which models the text-side prompts as latent variables to capture instance-level uncertainty; and (iii) *class-aware prior construction*, which regularizes the latent space using semantic information from both the input instance and its class prototype. An overview of the full framework is illustrated in Figure 1, which highlights the generation of text prompts from image features, the variational posterior, and the class-conditioned prior used for regularization.

## 4.1 Sample-specific Multi-Modal Prompt Generation

In multi-modal prompt learning, prior work typically learns a fixed set of prompt tokens shared across all input samples [5, 8]. While effective, such fixed prompts cannot adapt to instance-specific variations, which are crucial for robust downstream performance under distribution shifts.

To overcome this limitation, we propose *sample-specific prompt generation*, where prompts in the text encoder are dynamically generated based on the input image. Unlike previous methods that insert fixed prompts at a single layer, we generate and inject prompts across multiple transformer layers, enabling hierarchical and fine-grained modulation of the language stream.

Specifically, given an input image $x$, we first extract its global visual representation $f_x$ using the frozen CLIP image encoder. To generate text-side prompts, we define a set of $H$ layer-specific prompt generators $\{\Phi_i\}_{i=J}^{J+H-1}$, where each $\Phi_i$ is a lightweight MLP that maps the image feature to a set of $M$ prompt tokens for the $i$-th transformer layer:

$$z_i = \Phi_i(f_x) \in \mathbb{R}^{M \times d}, \quad i = J, \ldots, J + H - 1, \tag{12}$$

where $d$ is the token embedding dimension. Although all $\Phi_i$ share the same architecture, they have independent parameters to allow for layer-specific adaptation.

The resulting prompts $z_i$ are concatenated with the frozen text token embeddings $W_i$ at each layer $i$ of the CLIP text encoder, as formalized in Eq. 8, following the multi-layer prompt insertion strategy used in MaPLe [5] and MMRL [8].

In parallel, we also adopt deep vision prompting in the CLIP image encoder, where a set of learnable vision-side prompt tokens $\tilde{z}_{i-1} \in \mathbb{R}^{M \times d}$ is inserted at each selected transformer layer $i$. These vision prompts are shared across samples and optimized independently from the input $x$. They are not generated dynamically, nor modeled as latent variables.

In summary, our method introduces dynamic, sample-specific text prompts $z_i$ conditioned on image features and static, shared vision prompts $\tilde{z}_i$, aligning both modalities through a structured and hierarchical prompting design. Only the text-side prompts $z_i$ are modeled probabilistically in our variational framework.

## 4.2 Variational Multi-Modal Prompt Adaptation

While our sample-specific prompts $z_i$ increase flexibility, they remain deterministic and lack the ability to capture uncertainty–a key factor in few-shot and distribution-shift settings. To address this, we formulate prompt adaptation as a probabilistic latent variable model, replacing deterministic prompt tokens with latent prompt tokens learned via variational inference.

For each input image $x$ and fixed text template $t$, we define $z = \{z_i\}_{i=J}^{J+H-1}$, where each $z_i \in \mathbb{R}^{M \times d}$ is a latent variable representing the prompt tokens inserted at layer $i$ of the text encoder. To model the posterior distribution over these latent variables, we introduce a set of layer-specific MLPs $\{\phi_i\}_{i=J}^{J+H-1}$, where each $\phi_i$ predicts the parameters of a Gaussian distribution:

$$[\mu_i, \sigma_i] = \phi_i(\overline{f}_x), \tag{13}$$

where $\overline{f}_x$ is the frozen CLIP image embedding. Using these predicted parameters, the posterior distribution is formulated as a product of layer-wise Gaussians, conditioned solely on the image $x$:

$$q_\phi(z_i \mid x) = \mathcal{N}\left(\mu_i, \operatorname{diag}(\sigma_i^2)\right) \tag{14}$$

Given label $y$, we aim to maximize the marginal likelihood:

$$\log p(y \mid x, t) = \log \int p(y \mid x, t, z)\, p(z \mid x)\, dz. \tag{15}$$

As the integral is intractable, we maximize the variational evidence lower bound (ELBO):

$$\mathcal{L}_{\text{ELBO}} = \mathbb{E}_{q_\phi(z|x)}\left[\log p(y \mid x, t, z)\right] - \text{KL}\left(q_\phi(z \mid x) \,\|\, p(z)\right). \tag{16}$$

Here, $p(z)$ represents the prior distribution over the latent prompts, which we initially set to a standard Gaussian $\mathcal{N}(\mathbf{0}, \mathbf{I})$. In implementation, we apply the reparameterization trick [71] to enable gradient-based optimization:

$$z_i = \mu_i + \sigma_i \odot \epsilon_i, \quad \epsilon_i \sim \mathcal{N}(\mathbf{0}, \mathbf{I}). \tag{17}$$

Each sampled $z_i$ replaces the deterministic $z_i$, and is concatenated with the frozen tokens $w_i$ to form the layer input, as formalized in Eq. 8. This variational formulation introduces a structured, uncertainty-aware distribution over prompts and allows the model to adaptively control prompt behavior per input.

## 4.3 Class-Aware Prior Construction

In variational inference, the prior distribution $p(z)$ serves as a crucial regularizer for the learned posterior $q_\phi(z \mid x)$. While a standard choice $p(z) = \mathcal{N}(\mathbf{0}, \mathbf{I})$ provides a generic reference, it lacks semantic structure and offers no task-specific guidance. To incorporate class-level semantics into the latent prompt space, we introduce a *class-aware prior* that conditions on a prototype representation for each class.

During training, we assume access to the class label $y$ for each sample. We compute a class prototype $o_y$ by averaging the posterior means of training samples in class $y$:

$$o_y = \frac{1}{|D_y|} \sum_{x_i \in D_y} \overline{f}_{x_i}, \tag{18}$$

where $D_y$ is the set of training instances labeled $y$.

To model layer-wise latent prompts $z = \{z_i\}_{i=J}^{J+H-1}$, we introduce a set of layer-specific prior networks $\{\psi_i\}_{i=J}^{J+H-1}$. Each prior network maps the class prototype $c_y$ to the parameters of a Gaussian prior at layer $i$:

$$[\hat{\mu}_i, \hat{\sigma}_i] = \psi_i(c_y), \quad p_\psi(z_i \mid o_y) = \mathcal{N}(\hat{\mu}_i, \text{diag}((\hat{\sigma}_i)^2)). \tag{19}$$

The resulting ELBO objective now sums across layers:

$$\mathcal{L}_{\text{ELBO}} = \sum_{i=J}^{J+H-1} \left( \mathbb{E}_{q_\phi(z_i|x)}[\log p(y \mid x, t, z_i)] - \text{KL}\left(q_\phi(z_i \mid x) \,\|\, p_\psi(z_i \mid o_y)\right) \right). \tag{20}$$

This class-aware prior construction provides global semantic anchoring for each layer's prompt distribution. It encourages prompts from the same class to lie in nearby regions of the latent space, improving intra-class consistency and few-shot generalization. At test time, when $y$ is unavailable, we revert to the standard prior $p(z_i) = \mathcal{N}(\mathbf{0}, \mathbf{I})$ for all layers.

## 4.4 Inference Procedure

During inference, our method follows a single forward pass through the VaMP framework. Given an input image $x$ and a fixed text template $t$ (*e.g.*, "A photo of a [CLASS]"), we first extract the frozen CLIP image feature $\overline{f}_x$.

For probabilistic inference, we leverage Monte Carlo sampling over the latent prompt distribution. Specifically, given input image $x$, we draw $S$ samples $\{z_{i,s}\}_{s=1}^{S}$ from the learned posterior $q_\phi(z_i \mid x)$ for each layer $i$:

$$[\mu_i, \sigma_i] = \phi_i(\overline{f}_x), \quad z_{i,s} = \mu_i + \sigma_i \odot \epsilon_{i,s}, \quad \epsilon_{i,s} \sim \mathcal{N}(\mathbf{0}, \mathbf{I}). \tag{21}$$

Each sampled latent prompt $z_{i,s}$ is concatenated with the frozen text tokens $w_i$ at each transformer layer. Simultaneously, a set of shared vision-side prompts $\{\tilde{z}_i\}_{i=J}^{J+H-1}$ is injected into the corresponding layers of the CLIP image encoder. The model then computes the image and text features using the frozen CLIP encoders with inserted prompts, and generates a prediction $p_s(y \mid x, t, \{z_{i,s}\}_{i=J}^{J+H-1})$ based on the similarity between the projected image and text representations.

Finally, the predictions are averaged across samples to form the final output:

$$p(y \mid x, t) = \frac{1}{S} \sum_{s=1}^{S} p_s(y \mid x, t, \{z_{i,s}\}_{i=J}^{J+H-1}). \tag{22}$$

In our experiments, we use $S = 10$ samples for all evaluations. This inference-time ensembling preserves the expressiveness of the variational model while improving robustness and stability.

## 5 Experiments

### 5.1 Experiments Setup

We evaluate the performance of VaMP under three different settings: base-to-novel generalization, cross-dataset evaluation, domain generalization, and few-shot learning. All conducted under a 16-shot setting, where each category has only 16 training examples.

**Base-to-Novel Generalization.** In this setting, dataset classes are split into base and novel classes. The model is trained exclusively on base classes and tested on both base and novel classes, enabling an assessment of its transfer learning performance on base classes and its ability to preserve the inherent generalization and zero-shot capabilities of pre-trained VLMs for novel classes. This evaluation is conducted across 11 diverse classification datasets: ImageNet [72], Caltech101 [73], OxfordPets [74], StanfordCars [75], Flowers102 [76], Food101 [77], FGVCAircraft [78], SUN397 [79], UCF101 [80], DTD [81], and EuroSAT [82].

**Cross-Dataset Evaluation.** This evaluation examines the model's ability to generalize to unseen datasets. Following CoCoOp [3], the model is trained on all 1000 ImageNet classes in a few-shot setting and directly tested, without further fine-tuning, on the same datasets used for base-to-novel generalization, allowing us to assess cross-dataset transferability.

**Domain Generalization.** To evaluate the model's robustness to domain shifts and its generalization to out-of-distribution data, we train it on ImageNet and test it on four domain-variant datasets: ImageNetV2 [83], ImageNet-Sketch [84], ImageNet-A [85], and ImageNet-R [86], each introducing distinct types of domain variation.

**Implementation Details.** We follow prior studies [5, 8] and adopt a 16-shot training setting for all experiments unless otherwise noted. We build on the ViT-B/16 variant of CLIP [89] as the visual backbone and apply multi-layer prompt tuning on the text and vision encoders starting from the $J$-th transformer layer. For MMRL-style settings, we set $J = 5$, $H = 7$ and insert $M = 5$ learnable representation tokens per layer. For MaPLe-style configurations, we adopt $J = 0$, $H = 9$ with prompt length $M = 2$ for both modalities. All experiments are conducted on a single NVIDIA V100 GPU.

### 5.2 Main Results

**Base-to-Novel Generalization.** We evaluate VaMP against recent prompt tuning methods across 11 diverse datasets under the base-to-new generalization protocol. As shown in Table 1, VaMP achieves competitive performance on base classes while consistently outperforming all baselines on novel classes. In particular, VaMP attains the highest average novel accuracy of 78.67%, outperforming the best previous method (MMRL) by 1.51% and demonstrating strong generalization to unseen categories. The advantage is especially pronounced on challenging datasets with large domain shifts. For example, on DTD—a dataset characterized by fine-grained textures rather than semantic categories—VaMP achieves a novel accuracy of 75.50%, surpassing MMRL by 2.67%. These results highlight the strength of our structured, variational modeling in adapting to unfamiliar domains without sacrificing base class performance.

**Domain Generalization.** We further evaluate VaMP in a domain generalization setting,

**Table 1:** Comparison of VaMP with previous state-of-the-art methods on base-to-novel generalization across 11 datasets. Bold values indicate the best results. VaMP consistently enhances base class performance without compromising generalization.

| Method | Average | | | ImageNet | | | Caltech101 | | | OxfordPets | | |
|---|---|---|---|---|---|---|---|---|---|---|---|---|
| | Base | Novel | H | Base | Novel | H | Base | Novel | H | Base | Novel | H |
| CLIP [1] | 69.34 | 74.22 | 71.70 | 72.43 | 68.14 | 70.22 | 96.84 | 94.00 | 95.40 | 91.17 | 97.26 | 94.12 |
| CoOp [2] | 82.69 | 63.22 | 71.66 | 76.47 | 67.88 | 71.92 | 98.00 | 89.81 | 93.73 | 93.67 | 95.29 | 94.47 |
| CoOpOp [3] | 80.47 | 71.69 | 75.83 | 75.98 | 70.43 | 73.10 | 97.96 | 93.81 | 95.84 | 95.20 | 97.69 | 96.43 |
| ProDA [38] | 81.56 | 72.30 | 76.65 | 75.40 | 70.23 | 72.72 | 98.27 | 93.23 | 95.68 | 95.43 | 97.83 | 96.62 |
| KgCoOp [41] | 80.73 | 73.60 | 77.00 | 75.83 | 69.96 | 72.78 | 97.72 | 94.39 | 96.03 | 94.65 | 97.76 | 96.18 |
| MaPLe [5] | 82.28 | 75.14 | 78.55 | 76.66 | 70.54 | 73.47 | 97.74 | 94.36 | 96.02 | 95.43 | 97.76 | 96.58 |
| PromptSRC [6] | 84.26 | 76.10 | 79.97 | 77.60 | 70.73 | 74.01 | 98.10 | 94.03 | 96.02 | 95.33 | 97.30 | 96.30 |
| TCP [42] | 84.13 | 75.36 | 79.51 | 77.27 | 69.87 | 73.38 | 98.23 | 94.67 | 96.42 | 94.67 | 97.20 | 95.92 |
| MMA [70] | 83.20 | 76.80 | 79.87 | 77.31 | 71.00 | 74.02 | 98.40 | 94.00 | 96.15 | 95.40 | **98.07** | 96.72 |
| 2SFS [87] | 85.55 | 75.48 | 80.20 | 77.71 | 70.99 | 74.20 | 98.71 | 94.43 | 96.52 | 95.32 | 97.82 | 96.55 |
| SkipT [88] | 85.04 | 77.53 | 81.11 | 77.73 | 70.40 | 73.89 | 98.50 | 95.33 | 96.89 | 95.70 | 97.87 | **96.77** |
| MMRL [8] | 85.68 | 77.16 | 81.20 | 77.90 | 71.30 | 74.45 | **98.97** | 94.50 | 96.68 | 95.90 | 97.60 | 96.74 |
| VaMP | **86.45** | **78.67** | **82.37** | **78.98** | **73.45** | **76.11** | 98.95 | **95.96** | **97.43** | **96.95** | 95.24 | 96.08 |

| Method | StanfordCars | | | Flowers102 | | | Food101 | | | FGVCAircraft | | |
|---|---|---|---|---|---|---|---|---|---|---|---|---|
| | Base | Novel | H | Base | Novel | H | Base | Novel | H | Base | Novel | H |
| CLIP [1] | 63.37 | 74.89 | 68.65 | 72.08 | 77.80 | 74.83 | 90.10 | 91.22 | 90.66 | 27.19 | 36.29 | 31.09 |
| CoOp [2] | 78.12 | 60.40 | 68.13 | 97.60 | 59.67 | 74.06 | 88.33 | 82.26 | 85.19 | 40.44 | 22.30 | 28.75 |
| CoOpOp [3] | 70.49 | 73.59 | 72.01 | 94.87 | 71.75 | 81.71 | 90.70 | 91.29 | 90.99 | 33.41 | 23.71 | 27.74 |
| ProDA [38] | 74.70 | 71.20 | 72.91 | 97.70 | 68.68 | 80.66 | 90.30 | 88.57 | 89.43 | 36.90 | 34.13 | 35.46 |
| KgCoOp [41] | 71.76 | 75.04 | 73.36 | 95.00 | 74.73 | 83.65 | 90.50 | 91.70 | 91.09 | 36.21 | 33.55 | 34.83 |
| MaPLe [5] | 72.94 | 74.00 | 73.47 | 95.92 | 72.46 | 82.56 | 90.71 | 92.05 | 91.38 | 37.44 | 35.61 | 36.50 |
| PromptSRC [6] | 78.27 | 74.97 | 76.58 | 98.07 | 76.50 | 85.95 | 90.67 | 91.53 | 91.10 | 42.73 | 37.87 | 40.15 |
| TCP [42] | 80.80 | 74.13 | 77.32 | 97.73 | 75.57 | 85.23 | 90.57 | 91.37 | 90.97 | 41.97 | 34.43 | 37.83 |
| MMA [70] | 78.50 | 73.10 | 75.70 | 97.77 | 75.93 | 85.48 | 90.13 | 91.34 | 90.71 | 40.57 | 36.33 | 38.33 |
| 2SFS [87] | 82.50 | 74.80 | 78.46 | 98.29 | 76.17 | 85.83 | 89.11 | 91.34 | 90.21 | 47.48 | 35.51 | 40.63 |
| SkipT [88] | 82.93 | 72.50 | 77.37 | 98.57 | 75.80 | 85.70 | 90.67 | 92.03 | 91.34 | 45.37 | 37.13 | 40.84 |
| MMRL [8] | 81.30 | 75.07 | 78.06 | **98.97** | 77.27 | 86.78 | 90.57 | 91.50 | 91.03 | 46.30 | 37.03 | 41.15 |
| VaMP | 83.78 | 80.14 | 81.91 | 98.96 | 83.97 | 90.85 | 92.77 | 93.16 | 92.96 | 46.77 | 41.13 | 43.76 |

| Method | SUN397 | | | DTD | | | EuroSAT | | | UCF101 | | |
|---|---|---|---|---|---|---|---|---|---|---|---|---|
| | Base | Novel | H | Base | Novel | H | Base | Novel | H | Base | Novel | H |
| CLIP [1] | 69.36 | 75.35 | 72.23 | 53.24 | 59.90 | 56.37 | 56.48 | 64.05 | 60.03 | 70.53 | 77.50 | 73.85 |
| CoOp [2] | 80.60 | 65.89 | 72.51 | 79.44 | 41.18 | 54.24 | 92.19 | 54.74 | 68.69 | 84.69 | 56.05 | 67.46 |
| CoOpOp [3] | 79.74 | 76.86 | 78.27 | 77.01 | 56.00 | 64.85 | 87.49 | 60.04 | 71.21 | 82.33 | 73.45 | 77.64 |
| ProDA [38] | 78.67 | 76.93 | 77.79 | 80.67 | 56.48 | 66.44 | 83.90 | 66.00 | 73.88 | 85.23 | 71.97 | 78.04 |
| KgCoOp [41] | 80.29 | 76.53 | 78.36 | 77.55 | 54.99 | 64.35 | 85.64 | 64.34 | 73.48 | 82.89 | 76.67 | 79.65 |
| MaPLe [5] | 80.82 | 78.70 | 79.75 | 80.36 | 59.18 | 68.16 | 94.07 | 73.23 | 82.35 | 83.00 | 78.66 | 80.77 |
| PromptSRC [6] | 82.67 | 78.47 | 80.52 | 83.37 | 62.97 | 71.75 | 92.90 | 73.90 | 82.32 | 87.10 | 78.80 | 82.74 |
| TCP [42] | 82.63 | 78.20 | 80.35 | 82.77 | 58.07 | 68.25 | 91.63 | 74.73 | 82.32 | 87.13 | 80.77 | 83.83 |
| MMA [70] | 82.27 | 78.57 | 80.38 | 83.20 | 65.63 | 73.38 | 85.46 | **82.34** | 83.87 | 86.23 | 80.03 | 82.20 |
| 2SFS [87] | 82.59 | 78.91 | 80.70 | 84.60 | 65.01 | 73.52 | 96.91 | 67.09 | 79.29 | 87.85 | 78.19 | 82.74 |
| SkipT [88] | 82.40 | 79.03 | 80.68 | 83.77 | 67.23 | 74.59 | 92.47 | 83.00 | 87.48 | 87.30 | **82.47** | **84.81** |
| MMRL [8] | 83.20 | 79.30 | **81.20** | 85.67 | 65.00 | 73.82 | 95.60 | 80.17 | **87.21** | 88.10 | 80.07 | 83.89 |
| VaMP | **83.37** | 78.95 | 81.09 | **86.14** | 67.20 | 75.50 | 95.78 | 77.21 | 85.49 | **88.52** | 78.99 | 83.48 |

where prompts are optimized on ImageNet and directly applied to its variants (-V2, -S, -A, -R) without any target supervision. As shown in Table 2, VaMP achieves the highest accuracy across all four target domains, with an average accuracy of 61.73%, outperforming the best baseline (MMRL) by 1.20%. The gain is particularly significant on Sketch, a challenging domain due to its abstract and textureless nature. On this subset, VaMP achieves 49.69%, improving over MMRL by 0.52%. These results vali-

**Table 2:** Comparison of VaMP with previous state-of-the-art methods on domain generalization across 4 datasets.

| | Source | Target | | | |
|---|---|---|---|---|---|
| | ImageNet | -V2 | -S | -A | -R |
| CLIP [1] | 66.73 | 60.83 | 46.15 | 47.77 | 73.96 |
| CoOp [2] | 71.51 | 64.20 | 47.99 | 49.71 | 75.21 |
| CoOpOp [3] | 71.02 | 64.07 | 48.75 | 50.63 | 76.18 |
| MaPLe [5] | 70.72 | 64.07 | 49.15 | 50.90 | 76.98 |
| PromptSRC [6] | 71.27 | 64.35 | 49.55 | 50.90 | 77.80 |
| MMA [70] | 71.00 | 64.33 | 49.13 | 51.12 | 77.32 |
| MMRL [8] | 72.03 | 64.47 | 49.17 | 51.20 | 77.53 |
| VaMP | **72.83** | **64.96** | **49.69** | **51.97** | **78.01** |

date the effectiveness of our variational prompt modeling and class-aware regularization in enhancing out-of-distribution generalization, even in low-texture, cross-modality scenarios.

**Cross-Dataset Generalization.** To assess robustness under domain shift, we evaluate VaMP on cross-dataset transfer, where prompts are tuned on one source dataset (ImageNet) and evaluated directly on unseen target datasets. As shown in Table 3, VaMP achieves the best average performance across 10 diverse target datasets, with an average accuracy of 67.74%, outperforming the strongest baseline (MMRL) by 0.49%. The improvement is particularly notable on challenging datasets with large domain gaps. For example, on EuroSAT—a remote sensing dataset with significant visual discrepancy from natural images—VaMP achieves a target accuracy of 53.82%, improving over

**Table 3:** Comparison of VaMP with previous state-of-the-art methods on cross-dataset evaluation across 10 datasets.

| Method | Source | Target | | | | | | | | | | |
|---|---|---|---|---|---|---|---|---|---|---|---|---|
| | ImageNet | Average | Caltech101 | OxfordPets | StanfordCars | Flowers101 | Food101 | FGVCAircraft | SUN397 | DTD | EuroSAT | UCF101 |
| CoOp [2] | 71.51 | 63.88 | 93.70 | 89.14 | 64.51 | 68.71 | 85.30 | 18.47 | 64.15 | 41.92 | 46.39 | 66.55 |
| CoOpOp [3] | 71.02 | 65.74 | 94.43 | 90.14 | 65.32 | 71.88 | 86.06 | 22.94 | 67.36 | 45.73 | 45.37 | 68.21 |
| MaPLe [5] | 70.72 | 66.30 | 93.53 | 90.49 | 65.57 | 72.23 | 86.20 | 24.74 | 67.01 | 46.49 | 48.06 | 68.69 |
| PromptSRC [6] | 71.27 | 65.81 | 93.60 | 90.25 | 65.70 | 70.25 | 86.15 | 23.90 | 67.10 | **46.87** | 45.50 | 68.75 |
| TCP [42] | 71.40 | 66.29 | 93.97 | 91.25 | 64.69 | 71.21 | **86.69** | 23.45 | 67.15 | 44.35 | 51.45 | 68.73 |
| MMA [70] | 71.00 | 66.61 | 93.80 | 90.30 | **66.13** | 72.07 | 86.12 | 25.33 | **68.17** | 46.57 | 49.24 | 68.32 |
| MMRL [8] | 72.03 | 67.25 | 94.67 | 91.43 | 66.10 | 72.77 | 86.40 | 26.30 | 67.57 | 45.90 | 53.10 | 68.27 |
| VaMP | **72.83** | **67.74** | **94.96** | **91.79** | 66.10 | **73.18** | 86.97 | **26.76** | 68.04 | 46.82 | **53.82** | **68.93** |

**Table 4:** Ablation studies: impact of sample-specific multi-modal prompt generation, variational prompt adaptation and class-aware prior on base-to-new generalization performance, averaged across 11 datasets.

**(a) Effects of sample-specific multi-modal prompt generation**

| Method | Prompt Type | Base | New | H |
|---|---|---|---|---|
| MaPLe [5] | task-specific | 82.28 | 75.14 | 78.55 |
| | **sample-specific** | **82.95** | **76.95** | **79.83** |
| MMRL [8] | task-specific | 85.68 | 77.16 | 81.20 |
| | **sample-specific** | **85.93** | **78.13** | **81.84** |

**(b) Effects of variational multi-modal prompt generation**

| Method | Prompt Type | Base | New | H |
|---|---|---|---|---|
| MaPLe [5] | Deterministic prompt | 82.95 | 76.95 | 79.83 |
| | **Variational prompt** | **84.77** | **77.32** | **80.87** |
| MMRL [8] | Deterministic prompt | 85.93 | 78.13 | 81.84 |
| | **Variational prompt** | **86.11** | **78.45** | **82.10** |

**(c) Effects of class-aware prior** on base-to-new generalization

| Method | Prompt Type | Base | New | H |
|---|---|---|---|---|
| MaPLe [5] | Normal gaussian prior | 84.77 | 77.32 | 80.87 |
| | **Class-aware prior** | **85.13** | **78.07** | **81.45** |
| MMRL [8] | Normal gaussian prior | 86.11 | 78.45 | 82.10 |
| | **Class-aware prior** | **86.45** | **78.67** | **82.37** |

MMRL by 0.72%. These results demonstrate that our structured and uncertainty-aware adaptation generalizes well across domains, even when the target distribution differs substantially from the source.

## 5.3 Ablation Study

**Effects of Sample-specific Multi-Modal Prompt Generation.** We begin by assessing the effect of sample-specific multi-modal prompt generation. As shown in Table 4a, previous methods such as MaPLe and MMRL use task-specific prompts shared across all inputs. In contrast, our sample-specific design generates prompts conditioned on each input instance and injected across multiple layers. This structured, instance-aware formulation consistently improves generalization to novel categories, demonstrating the advantage of personalized multi-modal adaptation over fixed prompt configurations.

**Effects of Variational Prompt Adaptation.** Next, we evaluate the effect of variational modeling on the prompt tokens. While prior work adopts deterministic prompt embeddings, our approach treats the text-side prompts as latent variables inferred per instance. This formulation introduces uncertainty modeling into the prompt space, enabling better adaptation to ambiguous or out-of-distribution inputs. As shown in Table 4b, applying variational prompt learning improves generalization across both MaPLe and MMRL backbones, validating the benefit of latent, sample-specific modeling.

**Effects of Class-Aware Prior.** We further assess the role of structured priors in regularizing the latent prompt space. Unlike standard variational methods that use an isotropic Gaussian prior, our method constructs a class-aware prior from prototype representations computed over training data. This provides global semantic guidance for latent prompt inference. As shown in Table 4c, replacing the normal prior with a class-aware one consistently improves performance, highlighting the importance of semantic regularization for class-consistent prompt adaptation.

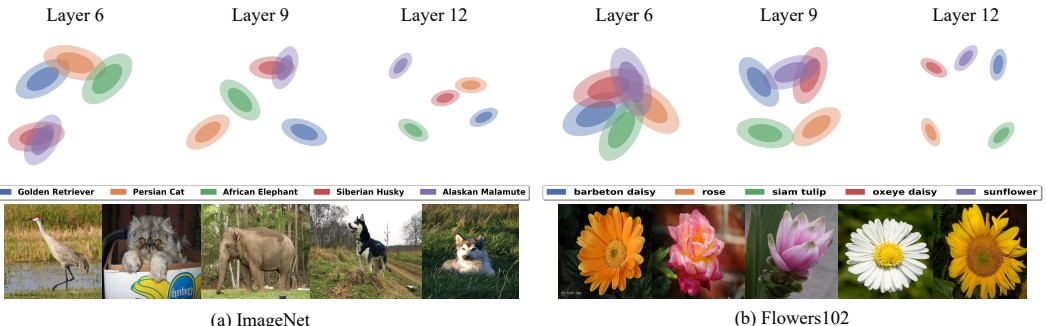

**Figure 2: Qualitative analysis.** Layer-wise visualization of aggregated posterior mean distributions $q_\phi(\mathbf{z}_i|x)$ for sample images from (a) ImageNet and (b) Flowers102.

## 5.4 Qualitative Analysis

To gain intuitive insights into the learned latent prompt space, we visualize the posterior distributions across text encoder layers. Figure 2 illustrates this process for representative samples from ImageNet and Oxford Flowers datasets. For each input image $x$, we extract the image-conditioned posterior $q_\phi(z_i|x)$ at layers 6, 9, and 12. Since VaMP generates $M$ prompt tokens per layer, we aggregate their means and covariances as $\mu_{\text{agg},i}(x) = \frac{1}{M} \sum_{j=1}^{M} \mu_i^j(x)$ and $\Sigma_{\text{agg},i}(x) = \frac{1}{M} \sum_{j=1}^{M} \Sigma_i^j(x)$, then project these high-dimensional vectors to 2D using PCA. Contours represent uncertainty regions at $1.5\sigma$ and $2.5\sigma$ standard deviations.

These visualizations provide insights into how VaMP achieves uncertainty-aware prompt learning through its variational design. First, the varying distributional characteristics across layers and samples—reflected in the spatial positions, orientations, and covariance structures of the posterior distributions—demonstrate that VaMP successfully models input-dependent uncertainty through its variational framework, moving beyond deterministic point estimates. Second, the layer-wise evolution reveals hierarchical uncertainty refinement: early layers (Layer 6) exhibit broader posterior distributions with larger variance to accommodate diverse prompt adaptations, while deeper layers (Layer 12) manifest tighter, more concentrated distributions with reduced uncertainty reflecting task-specific specialization. This progressive variance reduction enables adaptive representation learning at different semantic abstraction levels. Third, the consistent inter-class separation of posterior distributions across all depths demonstrates the effectiveness of the class-aware prior $p_\theta(z|c)$, which regularizes the variational posterior toward discriminative regions of the latent space. This class-conditional guidance ensures that prompts encode category-specific information from early feature extraction stages. Finally, VaMP's stochastic sampling mechanism, manifested through the distributional spread and overlap patterns, provides inherent robustness against overfitting to singular prompt configurations. By parameterizing prompts as probability distributions rather than fixed embeddings, VaMP balances representation flexibility with discriminative capacity throughout the hierarchical architecture.

## 6 Conclusion

We presented VaMP, a variational framework for prompt adaptation in multi-modal representation learning. Our approach addresses the limitations of fixed, shared prompts by introducing a structured and uncertainty-aware mechanism that adapts to individual input instances. VaMP comprises three key components: (*i*) sample-specific multi-modal prompt generation, where visual features condition prompt tokens across multiple transformer layers; (*ii*) variational modeling of text-side prompts as latent variables, enabling instance-specific and probabilistic adaptation; and (*iii*) a class-aware prior constructed from semantic prototypes, which regularizes the latent space with global class-level information. Through extensive experiments on few-shot and domain generalization benchmarks, we demonstrate that VaMP achieves state-of-the-art performance while maintaining high parameter efficiency. Our findings highlight the importance of modeling both instance-level variability and task structure in prompt-based adaptation for vision-language models.

## Acknowledgements

This work is supported by the Hong Kong Research Grants Council - General Research Fund (Grant No.: 17211024).

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

# Appendix

## A   Dataset Details

Details of 14 datasets are shown in Table A1.

**Table A1:** Summary of the 14 datasets.

| Dataset | Classes | Train | Val | Test | Description |
|---|---|---|---|---|---|
| ImageNet [72] | 1,000 | 1.28M | $\sim$ | 50,000 | Recognition of generic objects |
| Caltech101 [73] | 100 | 4,128 | 1,649 | 2,465 | Recognition of generic objects |
| OxfordPets [74] | 37 | 2,944 | 736 | 3,669 | Fine-grained classification of pets |
| StanfordCars [75] | 196 | 6,509 | 1,635 | 8,041 | Fine-grained classification of cars |
| Flowers102 [76] | 102 | 4,093 | 1,633 | 2463 | Fine-grained classification of flowers |
| Food101 [77] | 101 | 50,500 | 20,200 | 30,300 | Fine-grained classification of foods |
| FGVCAircraft [78] | 100 | 3,334 | 3,333 | 3,333 | Fine-grained classification of aircrafts |
| SUN397 [79] | 397 | 15,880 | 3,970 | 19,850 | Scene classification |
| DTD [81] | 47 | 2,820 | 1,128 | 1,692 | Texture classification |
| EuroSAT [82] | 10 | 13,500 | 5,400 | 8,100 | Land use & cover classification with satellite images |
| UCF101 [80] | 101 | 7,639 | 1,898 | 3,783 | Action recognition |
| ImageNetV2 [83] | 1,000 | $\sim$ | $\sim$ | 10,000 | New test data for ImageNet |
| ImageNet-Sketch [84] | 1,000 | $\sim$ | $\sim$ | 50,889 | Sketch-style images of ImageNet classes |
| ImageNet-A [85] | 200 | $\sim$ | $\sim$ | 7,500 | Natural adversarial examples of 200 ImageNet classes |
| ImageNet-R [86] | 200 | $\sim$ | $\sim$ | 30,000 | Renditions of 200 ImageNet classes |

## B   More Implementation Details

All models are trained using the AdamW optimizer with a learning rate of 0.001 and a weight decay of 0.01. The batch size is set to 32 for ImageNet and 4 for all other datasets. We apply automatic mixed-precision training throughout to improve efficiency. For base-to-novel generalization on ImageNet, we train for 5 epochs; for other datasets we train for 10 epochs. For cross-dataset and domain generalization tasks, we train on ImageNet for a single epoch. Few-shot learning tasks use 5 training epochs on ImageNet and 50 epochs on target datasets. All reported results are averaged over three independent runs. All prompts and representation tokens are initialized from a zero-mean Gaussian distribution with a standard deviation of 0.02. For EuroSAT, we follow MMRL [8] and set the representation token dimension $d_r = 2048$; for all other datasets, we use $d_r = 512$. The fusion parameter $\alpha$ in MMRL-style classifiers is fixed to 0.7. The average accuracy is reported over three independent runs. For variational modeling, we use a two-layer MLP with GELU activation to parameterize both the posterior network $\phi$ and the prior network $\psi$, outputting mean and log-variance vectors per layer. The latent variables $z$ are sampled using the reparameterization trick, and we perform $S = 10$ Monte Carlo samples at inference time. Class prototypes $o_y$ are computed offline at the start of training.

## C   Efficiency Analysis

We analyze the efficiency of our variational multi-modal prompt framework, focusing on the trade-off between inference cost and generalization ability. As detailed in Table A2, VaMP introduces only a modest overhead compared to the strong MMRL baseline. For instance, with $S = 10$, our method improves the Harmonic Mean (HM) from 81.20 to 82.37 with a minimal latency increase of just 0.8ms per image on an NVIDIA V100 GPU. The performance gains saturate quickly as $S$ increases, demonstrating that VaMP achieves an effective balance between accuracy and efficiency even with limited sampling. Furthermore, our approach is parameter-efficient, increasing the number of learnable parameters by less than 6% when integrated into MMRL (from 4.992M to 5.132M). This balance demonstrates that VaMP can be deployed with low overhead while still leveraging uncertainty modeling for improved generalization.

**Table A2:** Analysis of performance vs. efficiency trade-off.

| Method | S | Inference Time (ms/img) | Base | Novel | HM |
|---|---|---|---|---|---|
| MMRL | - | 5.3 | 85.68 | 77.16 | 81.20 |
| MMRL+VaMP | 1 | 5.5 | 86.10 | 77.35 | 81.44 |
| MMRL+VaMP | 5 | 5.8 | 86.37 | 78.26 | 82.03 |
| MMRL+VaMP | 10 | 6.1 | **86.45** | **78.67** | **82.37** |
| MMRL+VaMP | 20 | 6.5 | **86.47** | 78.37 | 82.22 |
| MMRL+VaMP | 50 | 9.3 | 86.43 | 78.15 | 82.01 |

# D   More Ablation Studies

We conduct ablation studies by integrating our variational prompt adaptation into the MMRL [8] framework, analyzing the impact of key hyperparameters such as the prompt depth, and prompt length. All results are reported on the base-to-novel generalization benchmark (11 datasets), using average accuracy across base and novel splits.

**Effect of Prompt Insertion Depth $J$ and $H$.**   We vary the transformer layer index $J$ where prompt tuning begins, and the number of consecutive layers $H$ to which prompts are added. As shown in Table A3, inserting prompts deeper into the encoder (e.g., starting from layer $J = 5$) and extending them to more layers (e.g., $H = 7$) leads to better performance on both base and novel classes. This confirms the benefit of hierarchical prompt injection for deeper vision-language alignment.

**Table A3:** Effect of prompt insertion depth $(J, H)$.

| $J$ | $H$ | Base | Novel | H |
|---|---|---|---|---|
| 4 | 3 | 85.57 | 77.14 | 81.13 |
| 6 | 5 | 86.28 | 78.02 | 82.01 |
| 6 | 7 | **86.45** | **78.67** | **82.37** |

**Effect of Prompt Token Length $M$.**   We analyze the sensitivity to the number of prompt tokens $M$ injected per layer. Table A4 shows that increasing $M$ from 1 to 5 improves accuracy, as the model benefits from higher representational capacity. However, further increasing to $M = 8$ slightly reduces generalization, likely due to redundancy and overfitting. Hence, we set $M = 5$ as the default in our main experiments.

**Table A4:** Effect of prompt token length $M$ per layer.

| $M$ | Base | Novel | H |
|---|---|---|---|
| 1 | 85.76 | 77.21 | 81.30 |
| 3 | 86.18 | 78.05 | 82.02 |
| 5 | **86.45** | **78.67** | **82.37** |
| 8 | 86.19 | 78.12 | 82.00 |

# E   Detailed Derivation of the Variational Lower Bound (ELBO)

We derive the evidence lower bound (ELBO) used in our variational prompt learning framework. Our goal is to estimate the conditional likelihood of the class label $y$ given an image $x$ and a text prompt template $t$, where the latent variables $z$ represent the sample-specific text prompt tokens injected across multiple layers of the language encoder.

The marginal likelihood is obtained by integrating out the latent variables $z$:

$$\log p(y \mid x, t) = \log \int p(y \mid x, t, z) \, p(z \mid x) \, dz. \tag{23}$$

Since this integral is generally intractable, we introduce a variational posterior $q_\phi(z \mid x)$ and apply Jensen's inequality:

$$\log p(y \mid x, t) = \log \int q_\phi(z \mid x) \cdot \frac{p(y \mid x, t, z)\, p(z \mid x)}{q_\phi(z \mid x)}\, dz \tag{24}$$

$$\geq \mathbb{E}_{q_\phi(z|x)}\left[\log \frac{p(y \mid x, t, z)\, p(z \mid x)}{q_\phi(z \mid x)}\right] \tag{25}$$

$$= \underbrace{\mathbb{E}_{q_\phi(z|x)}[\log p(y \mid x, t, z)]}_{\text{expected log-likelihood}} - \underbrace{\mathrm{KL}(q_\phi(z \mid x) \,\|\, p(z \mid x))}_{\text{KL divergence}}. \tag{26}$$

We model $z$ as a collection of layer-wise latent prompt embeddings $\{z_i\}_{i=J}^{J+H-1}$, one for each of the $H$ transformer layers in the text encoder. We assume the posterior and prior factorize across layers:

$$q_\phi(z \mid x) = \prod_{i=J}^{J+H-1} q_\phi(z_i \mid x), \tag{27}$$

$$p(z \mid x) = \prod_{i=J}^{J+H-1} p(z_i \mid x). \tag{28}$$

This leads to the following form of the ELBO:

$$\mathcal{L}_{\text{ELBO}} = \sum_{i=J}^{J+H-1} \left[ \mathbb{E}_{q_\phi(z_i|x)} \log p(y \mid x, t, z_i) - \mathrm{KL}(q_\phi(z_i \mid x) \,\|\, p(z_i \mid x)) \right], \tag{29}$$

where $z_i$ is injected at layer $i$ of the frozen text encoder, modulating the token representations via concatenation.

During training, we replace $p(z_i \mid x)$ with a class-aware prior:

$$p_\psi(z_i \mid o_y) = \mathcal{N}(\hat{\mu}_i, \mathrm{diag}((\hat{\sigma}_i)^2)) \quad [\hat{\mu}_i, \hat{\sigma}_i] = \psi_i(c_y). \tag{30}$$

where the class prototype $o_y$ is the mean of posterior means $\hat{\mu}_i$ over all training samples in class $y$.

The final training objective maximizes the ELBO in Eq. 29.

This variational formulation enables our model to learn expressive, uncertainty-aware, sample-specific prompts while regularizing the latent space with class-level semantic structure.

## F   Extension to Other Tasks

To assess the generalizability of VaMP beyond standard image classification, we extended our evaluation to the more complex domains of open-vocabulary segmentation and action recognition. Our approach was integrated into two state-of-the-art frameworks: CAT-Seg [90] for segmentation and FROSTER [91] for action recognition, both of which utilize a CLIP ViT-B/16 backbone. This compatibility allowed for a seamless and direct evaluation of VaMP's effectiveness in these diverse tasks.

For the open-vocabulary segmentation task, we adhered to the established CAT-Seg protocol. The model was trained on the COCO-Stuff dataset (118K images, 171 categories) and subsequently evaluated on several challenging benchmarks, including ADE20K, PASCAL-Context, and PASCAL VOC, which feature a wide range of category scales (59-847 classes). The results are detailed in Table A5.

For open-vocabulary action recognition, we adopted the base-to-novel evaluation protocol from FROSTER. This setup involves training the model exclusively on the base classes from two widely used video benchmarks, Kinetics-400 and UCF-101. The primary metric is the model's ability to generalize to novel, unseen action categories during testing. Table A6 summarizes these results.

The consistent performance gains observed across both segmentation and action recognition tasks underscore the scalability and robust generalization capabilities of our proposed framework.

**Table A5:** Performance for open-vocabulary segmentation.

| Method | Prompt Tuning | ADE-847 | PC-459 | ADE-150 | PC-59 | VOC-20 |
|---|---|---|---|---|---|---|
| ZegFormer [92] | - | 5.6 | 10.4 | 18.0 | 45.5 | 89.5 |
| OVSeg [93] | - | 7.1 | 11.0 | 24.8 | 53.3 | 92.6 |
| SAN [94] | - | 10.1 | 12.6 | 27.5 | 53.8 | 94.0 |
| CAT-Seg [90] | - | 12.0 | 19.0 | 31.8 | 57.5 | 94.6 |
| CAT-Seg | MMRL | 12.8 | 18.7 | 32.4 | 57.9 | 94.3 |
| CAT-Seg | MMRL+VaMP | **13.9** (+1.1) | **20.3** (+1.6) | **33.3** (+0.9) | **58.6** (+0.7) | **95.2** (+0.9) |

**Table A6:** Performance for open-vocabulary action recognition. B: Base classes, N: Novel classes, HM: Harmonic mean.

| Method | Prompt Tuning | K400(B) | K400(N) | K400(HM) | UCF(B) | UCF(N) | UCF(HM) |
|---|---|---|---|---|---|---|---|
| ViFi-CLIP [95] | - | 76.4 | 61.1 | 67.9 | 92.9 | 67.7 | 78.3 |
| Open-VCLIP [96] | - | 76.5 | 62.6 | 68.9 | 94.8 | 77.5 | 85.3 |
| FROSTER [91] | - | 77.8 | 64.3 | 70.4 | 95.3 | 80.0 | 87.0 |
| FROSTER | MMRL | 78.3 | 64.1 | 70.5 | 95.5 | 80.2 | 87.2 |
| FROSTER | MMRL+VaMP | **78.8** (+0.5) | **64.8** (+0.7) | **71.1** (+0.6) | **96.1** (+0.6) | **81.0** (+0.8) | **87.9** (+0.7) |

# G   Scalability to Other VLM Architectures

To demonstrate the architecture-agnostic nature of our method, we validated VaMP on several state-of-the-art VLMs that represent the latest advances in CLIP-style architectures. Our method can be readily integrated into more complex VLMs, as it only requires access to intermediate transformer layers and the ability to inject prompt tokens. The variational adaptation module is lightweight (MLPs per layer) and does not assume any specific backbone structure. We therefore integrated VaMP into several prominent models, including EVA-CLIP [97], SigLIP [98], and SigLIP 2 [99]. These models share similar structural designs with CLIP, which enabled their rapid adaptation for our experiments. As shown in Table A7, VaMP obtains clear and consistent performance improvements across all VLMs, demonstrating its strong generalization capability to unseen categories—a critical challenge in prompt learning. The consistent improvements across different VLMs further substantiate the effectiveness and generalizability of our approach to different architectures.

**Table A7:** Performance comparison of different VLMs on ViT-B/16 backbone under base-to-novel generalization setting.

| VLM | Method | Base | Novel | H |
|---|---|---|---|---|
| CLIP [1] | MMRL | 85.68 | 77.16 | 81.20 |
| CLIP [1] | MMRL+VaMP | **86.45** (+0.77) | **78.67** (+1.51) | **82.37** (+1.17) |
| EVA-CLIP [97] | MMRL | 85.97 | 77.69 | 81.62 |
| EVA-CLIP [97] | MMRL+VaMP | **86.59** (+0.62) | **79.18** (+1.49) | **82.71** (+1.09) |
| SigLIP [98] | MMRL | 86.12 | 78.21 | 81.97 |
| SigLIP [98] | MMRL+VaMP | **86.88** (+0.76) | **79.45** (+1.24) | **83.00** (+1.03) |
| SigLIP 2 [99] | MMRL | 86.64 | 78.97 | 82.62 |
| SigLIP 2 [99] | MMRL+VaMP | **87.09** (+0.45) | **80.02** (+1.05) | **83.40** (+0.78) |

# H   Broader Impact and Limitations

Our work presents a variational framework for sample-specific, uncertainty-aware prompt adaptation in vision-language models, aiming to improve robustness under distribution shifts and limited supervision. The proposed method has the potential to benefit a wide range of downstream applications where multi-modal understanding and generalization are essential, such as assistive AI systems, open-world recognition, or low-resource domain transfer. The probabilistic modeling component can further inspire future efforts in calibrated and interpretable multi-modal adaptation. At present, we have not identified any ethical concerns associated with the real-world applications of this technology. However, we strongly recommend continuous monitoring and evaluation to ensure its responsible

and ethical deployment. Our approach also has certain limitations. First, our class-aware prior construction relies on access to class prototypes computed from training data, which may not be available in zero-label scenarios. Extending our method to work under fully unsupervised or few-label conditions remains an open direction. Second, our experiments focus on classification tasks; the extension to generative or structured prediction settings (*e.g.*, image captioning, VQA) requires further investigation and architectural adaptation. Despite these challenges, we believe our framework takes an important step toward principled and efficient multi-modal prompt learning, and hope it provides useful insights for future research in uncertainty-aware adaptation, vision-language alignment, and lightweight tuning strategies.

