# OpenReview forum: "VaMP: Variational Multi-Modal Prompt Learning for Vision-Language Models"
_NeurIPS.cc/2025/Conference — NeurIPS 2025 poster_

### Official Review · Reviewer_47nX · 2025-06-04

**Clarity:** 3
**Significance:** 2
**Originality:** 2
**Rating:** 3
**Confidence:** 3

**Summary:**

This paper transforms deterministic multimodal prompts into statistical ones to capture instance-level variations and model uncertainty, achieving performance that surpasses state-of-the-art methods across multiple settings and benchmarks.

**Questions:**

Please refer to the Weaknesses section above.

**Ethical Concerns:**

["NO or VERY MINOR ethics concerns only"]

**Final Justification:**

I have carefully reviewed comments and considered the feedback from the other reviewers, especially Reviewer VFp2. I remain concerned about the novelty of this work. Furthermore, after a deeper investigation into recent advances in this field, I found that the core idea of this work appears to be quite similar to that of [1].

[1]. Variational prompt tuning improves generalization of vision-language models, 2022

**Limitations:**

yes

**Quality:**

3

**Strengths And Weaknesses:**

Strengths:

1.The writing is clear and easy to follow.

2.The mathematical formulations are well-defined, and the reasoning is sound.

3.The method demonstrates superior performance, surpassing state-of-the-art prompting strategies.

Weaknesses:

1.I feel that uncertainty-based modeling may increase computational overhead. Could the authors compare the models and computational complexity of deterministic and uncertainty-based prompting strategies?

2.The current benchmarks and settings are relatively simple. Can the proposed method be extended to dense prediction or video-based benchmarks?

3.There are many strategies for uncertainty modeling, such as Monte Carlo sampling and evidence learning. Why did the authors choose to model the Gaussian mean and variance using MLPs?

4.According to Figure 1, the prompts only operate at the encoding stage. Why not consider integrating them at the decoding stage as well?

---

> ### Author Rebuttal · Authors · 2025-07-30
>
> We appreciate the reviewer's valuable feedback and recognition of VaMP's clear presentation, well-defined mathematical formulations, sound reasoning, and superior performance across multiple benchmarks. We carefully address the reviewer's comments below.
>
> > W1: I feel that uncertainty-based modeling may increase computational overhead. Could the authors compare the models and computational complexity of deterministic and uncertainty-based prompting strategies?
>
> **R:** We thank the reviewer for raising this important practical consideration. As mentioned in Section S3 of our supplementary material, we have provided a quantitative analysis of computational efficiency. Here, we further provide a more detailed quantitative analysis of the trade-off between inference cost and accuracy under different numbers of Monte Carlo samples S. As shown in Table A7, even with a modest number of samples (e.g., S=5 or S=10), VaMP achieves clear gains in harmonic mean accuracy over the strong MMRL baseline, with only marginal increases in inference time (e.g., 5.3ms/image → 6.1ms/image). Specifically, at S=10, VaMP improves HM from 81.20 to 82.37 with a latency increase of only 0.8ms per image—demonstrating an efficient balance between generalization and overhead. Notably, the performance saturates quickly with increasing S, suggesting that our method remains highly efficient even with limited sampling. This confirms that VaMP can be deployed with low overhead while still leveraging uncertainty modeling for improved generalization.
>
> **Table A7: Comparison of inference cost and parameter count.**
>
> | Method | S | Inference Time (ms/img) | Base | Novel | HM |
> |--------|---|------------------------|------|-------|-----|
> | MMRL | - | **5.3** | 85.68 | 77.16 | 81.20 |
> | MMRL+VaMP | 1 | 5.5 | 86.10 | 77.35 | 81.44 |
> | MMRL+VaMP | 5 | 5.8 | 86.37 | 78.26 | 82.03 |
> | MMRL+VaMP | 10 | 6.1 | 86.45 | **78.67** | **82.37** |
> | MMRL+VaMP | 20 | 6.5 | **86.47** | 78.37 | 82.22 |
> | MMRL+VaMP | 50 | 9.3 | 86.43 | 78.15 | 82.01 |
>
> > W2: The current benchmarks and settings are relatively simple. Can the proposed method be extended to dense prediction or video-based benchmarks?
>
> **R:** Thank you for this valuable suggestion. We evaluated VaMP's generalizability beyond classification by conducting experiments on open-vocabulary segmentation and action recognition tasks. We integrated VaMP into state-of-the-art frameworks of two tasks: CAT-Seg [a] for segmentation and FROSTER [b] for action recognition, utilizing CLIP ViT-B/16 as the backbone encoder. Both baselines are extensions built upon CLIP, allowing our method to be seamlessly integrated into these approaches.
>
> For open-vocabulary segmentation, we followed CAT-Seg's protocol, training on COCO-Stuff (118K images, 171 categories) and evaluating across multiple benchmarks including ADE20K, PASCAL-Context, and PASCAL VOC with varying category scales (59-847 classes). The results are shown in Table A8:
>
> **Table A8: Performance comparison of different methods on ViT-B/16 backbone for open-vocabulary segmentation.**
>
> | Method | Prompt Tuning | ADE-847 | PC-459 | ADE-150 | PC-59 | VOC-20 |
> |--------|---------------|---------|--------|---------|-------|--------|
> | ZegFormer | - | 5.6 | 10.4 | 18.0 | 45.5 | 89.5 |
> | OVSeg | - | 7.1 | 11.0 | 24.8 | 53.3 | 92.6 |
> | SAN | - | 10.1 | 12.6 | 27.5 | 53.8 | 94.0 |
> | CAT-Seg | - | 12.0 | 19.0 | 31.8 | 57.5 | 94.6 |
> | CAT-Seg | MMRL | 12.8 | 18.7 | 32.4 | 57.9 | 94.3 |
> | CAT-Seg | MMRL+VaMP | **13.9** (+1.1) | **20.3** (+1.6) | **33.3** (+0.9) | **58.6** (+0.7) | **95.2** (+0.9) |
>
> For open-vocabulary action recognition, we adopted FROSTER's base-to-novel evaluation protocol on Kinetics-400 and UCF-101, where models trained exclusively on base classes are required to generalize to unseen novel action categories. The results are shown in Table A9:
>
> **Table A9: Performance comparison under base-to-novel setting on two widely-used video benchmarks, i.e., Kinetics-400 and UCF-101. B: Base classes, N: Novel classes, HM: Harmonic mean.**
>
> | Method | Prompt Tuning | K400(B) | K400(N) | K400(HM) | UCF(B) | UCF(N) | UCF(HM) |
> |--------|---------------|---------|---------|----------|--------|--------|---------|
> | ViFi-CLIP | - | 76.4 | 61.1 | 67.9 | 92.9 | 67.7 | 78.3 |
> | Open-VCLIP | - | 76.5 | 62.6 | 68.9 | 94.8 | 77.5 | 85.3 |
> | FROSTER | - | 77.8 | 64.3 | 70.4 | 95.3 | 80.0 | 87.0 |
> | FROSTER | MMRL | 78.3 | 64.1 | 70.5 | 95.5 | 80.2 | 87.2 |
> | FROSTER | MMRL+VaMP | **78.8** (+0.5) | **64.8** (+0.7) | **71.1** (+0.6) | **96.1** (+0.6) | **81.0** (+0.8) | **87.9** (+0.7) |
>
> As shown, VaMP achieves consistent improvements across different tasks, demonstrating the scalability of our approach and its strong generalization capabilities.
>
> [a] CAT-Seg: Cost Aggregation for Open-Vocabulary Semantic Segmentation. CVPR 2024
>
> [b] FROSTER: Frozen CLIP Is A Strong Teacher for Open-Vocabulary Action Recognition. ICLR 2024.
>
> > W3: There are many strategies for uncertainty modeling, such as Monte Carlo sampling and evidence learning. Why did the authors choose to model the Gaussian mean and variance using MLPs?
>
> **R:** We adopt MLPs to model the posterior mean and variance due to their simplicity, efficiency, and compatibility with variational inference. This design choice follows the well-established principles of Variational Autoencoders (VAEs)[c], which have been widely adopted in the machine learning community for modeling uncertainty. Similar to VAEs, our approach allows lightweight, layer-wise modeling of uncertainty and integrates naturally with the reparameterization trick for stable end-to-end training. Compared to alternatives like evidence learning, our approach does not require additional supervision or calibration loss, and directly optimizes downstream performance. Empirically, Table 4(b) in our main manuscript shows consistent improvements over deterministic prompts, validating our choice. Exploring richer uncertainty modeling strategies remains a promising direction for future work.
>
> [c] Auto-Encoding Variational Bayes. ICLR 2014.
>
> > W4: According to Figure 1, the prompts only operate at the encoding stage. Why not consider integrating them at the decoding stage as well?
>
> **R:**  We would like to clarify that, our method builds on CLIP-like architectures following previous methods [79, 80, 24, 71, 11, 66, 15], which employ a dual encoder architecture—a vision encoder and a text encoder—without a decoding stage. As such, there is no separate decoder module where prompts could be inserted. All computations are encoder-based, and predictions are made via similarity in the joint embedding space. Our prompt adaptation targets the transformer layers within the text encoder, which directly influence the representation used for image-text matching. For specific models or applications when the decoders are needed, our prompt design can be equally applicable.

---

> > ### Comment · Reviewer_47nX · 2025-08-03
> >
> > Thank you for your response. I have carefully reviewed your comments and considered the feedback from the other reviewers, especially Reviewer VFp2. I remain concerned about the novelty of this work. Furthermore, after a deeper investigation into recent advances in this field, I found that the core idea of this work appears to be quite similar to that of [1]. I’m sorry, but I may not be able to maintain a positive recommendation at this time.
> >
> > [1]. Variational prompt tuning improves generalization of vision-language models, 2022

---

> > > ### Author Response · Authors · 2025-08-03
> > >
> > > Dear Reviewer 47nX,
> > >
> > > Thank you for your careful consideration of our work and for bringing up your concerns about novelty in relation to \[1]. We appreciate the opportunity to clarify the significant differences between our VaMP approach and the variational prompt tuning work in \[1].
> > >
> > > First, we would like to clarify that our paper already discusses Bayesian prompt learning and references \[1] in Lines 31–36 and Lines 77-83, acknowledging its contributions while also articulating our distinct innovations.
> > >
> > > Our VaMP framework differs fundamentally from \[1] in several key aspects:
> > >
> > > 1. **Multi-modal vs. Single-modal Variational Prompting**:
> > >    VaMP is designed for multi-modal prompt learning in vision-language models, where variational modeling is applied to the textual prompt tokens to adaptively modulate both visual and textual representations. This design supports cross-modal alignment while maintaining architectural modularity. In contrast, \[1] focuses solely on the language modality and does not consider multi-modal alignment—an essential factor in vision-language generalization.
> > >
> > > 2. **Hierarchical Token-level Prompt Modeling**:
> > >    While \[1] applies variational inference at the *input-level* textual prompt only, VaMP introduces *token-wise variational modeling* across multiple transformer layers in both modalities. This hierarchical structure enables finer-grained, layer-aware modulation of internal representations, offering more flexible and expressive adaptation.
> > >
> > > 3. **Task- and Class-aware Prior Construction**:
> > >    VaMP incorporates semantic priors that integrate both instance features and class prototypes to capture task- and class-level structure. This structured prior promotes better generalization. In contrast, \[1] adopts a standard Gaussian prior without such semantic conditioning.
> > >
> > > 4. **Performance Improvements**:
> > >    Empirically, our method outperforms \[1] consistently. Specifically, VaMP achieves **+3.73%** improvement on *novel classes*, **+1.79%** on *cross-dataset generalization*, and **+0.72%** on *domain generalization* benchmarks.
> > >
> > > We believe these architectural and empirical distinctions clearly demonstrate that VaMP goes far beyond \[1] in both design and effectiveness, establishing its novelty and contribution in the context of multi-modal generalization.
> > >
> > > Please let us know if further clarification would be helpful.
> > >
> > > Best regards,
> > >
> > > Authors of Submission6566

---

> > > > ### Comment · Reviewer_47nX · 2025-08-04
> > > >
> > > > Thank you to the authors and Reviewer EaPS for the responses. My views are as follows:
> > > >
> > > > 1.The core of this work, namely variational inference on prompts, is essentially consistent with [1]. The overall contribution is incremental. Although the authors attempt to extend it to the multimodal setting, the insight, which is conditioned prompting, is rather trivial.
> > > >
> > > > 2.Regarding token-level prompt modeling and prior-guided strategies, these approaches are not novel and have already been explored in related work within this field.
> > > >
> > > > Therefore, I remain concerned about the originality of the work.

---

> ### Author Response · Authors · 2025-08-04
>
> Thank you for the reply. We would like to clarify several important points regarding the originality and contributions of our work.
>
> **Regarding Q1**
>
> We hold a sincere belief in the importance of evaluating research novelties and contributions within their rightful context and problem scope. We would like to clarify that, both [1] and VaMP build upon foundational techniques from established variational inference literature [a, b]. Indeed, [1] also clearly pointed out that their variational inference modeling was inspired by these works (P4, Sec 3.2).
>
> Below, we will clarify the key differences between VaMP and [1]:
>
> [1] applies variational inference techniques **only to the input textual prompts** by modeling them as latent variables drawn from **a standard Gaussian distribution**, enabling diverse prompt generation during inference for improved robustness.
>
> However, this approach has several limitations. First, by restricting variational modeling to **input-level prompts with global latent variables**, [1] cannot capture hierarchical feature interactions throughout the network nor fine-grained semantic variations between individual tokens, significantly limiting representational capacity. Additionally, [1]'s **text-only prompt tuning** disregards valuable information from visual modalities that could better align cross-modal representations. Finally, **the standard Gaussian prior** assumption is shared across all classes, preventing the model from capturing inter-class variations and resulting in less discriminative prompt distributions.
>
> In contrast, VaMP addresses these weaknesses through a comprehensive variational framework that operates across modalities and network depths. By implementing **token-wise variational modeling across multiple intermediate network layers**, VaMP simultaneously captures fine-grained semantic relationships between individual tokens and enables prompts to influence representations at multiple abstraction levels throughout the network hierarchy. As evidenced in Table 1, our model performs quite well on fine-grained datasets, with consistent improvements across base classes, novel classes, and harmonic mean metrics compared to SOTA approaches. Particularly notable gains are observed on StanfordCars (Base +2.48, Novel +5.07, H +3.85), Food101 (Base +2.20, Novel +1.66, H +1.93), FGVCAircraft (Base +0.47, Novel +4.10, H +2.61), and DTD (Base +0.47, Novel +2.20, H +1.68). These results further demonstrate the importance of both token-wise and layer-wise variational modeling for capturing subtle discriminative features in fine-grained visual classification tasks. Secondly, VaMP's **multi-modal design** incorporates both visual and textual signals when inferring posterior distributions, creating more aligned cross-modal representations. Finally, by employing **class-aware priors** instead of standard Gaussian distributions, VaMP generates more discriminative prompts that better capture category-specific features and decision boundaries.
>
> Therefore, we believe that it is not a fair judgment to say that “the insight … is rather trivial”. Our idea and design principles are fundamentally different from [1].
>
> [a] Diederik P Kingma and Max Welling. Auto-encoding variational bayes. ICLR 2014
>
> [b] Jonathan Gordon, John Bronskill, Matthias Bauer, Sebastian Nowozin, and Richard Turner. Meta-learning probabilistic inference for prediction. ICLR 2019.
>
> **Regarding Q2**
>
> - Regarding token-level prompt modeling, existing approaches[24, 15] treat tokens as **deterministic optimization targets** that cannot capture uncertainty or adaptively modulate based on cross-modal alignment confidence. However, our method **treats each token as a latent variable under a probabilistic variational framework**, rather than a deterministic optimization target. This uncertainty-aware token modulation allows us to capture ambiguity in the cross-modal alignment and selectively adapt informative tokens. This probabilistic formulation is novel and provides clear performance and generalization benefits over deterministic tuning strategies, as shown in Table 4.
>
> - Regarding prior-guided strategies, existing variational prompt tuning methods[1] universally adopt **standard Gaussian distributions** as priors, while we are the first to introduce **class-aware prototypes** that integrate both instance features and class prototypes as priors to capture task- and class-level structure in variational prompt tuning. This class-aware prior design enables the model to leverage semantic relationships between categories and provides more informed initialization for prompt tokens, leading to better convergence and stronger cross-modal alignment compared to uninformative standard Gaussian priors.
>
> We sincerely appreciate the reviewer's comments. We will further clarify these distinctions in our revision. Any further comments and suggestions are welcome.

---

> > ### Author Response · Authors · 2025-08-07
> >
> > Dear Reviewer 47nX,
> >
> > Thank you for the reviewer's time and valuable feedback during the discussion period. As we approach the end of this period, we want to proactively reach out in case any additional clarification might be helpful before the deadline.
> >
> > Specifically, we would like to confirm whether our previous response adequately addressed the concerns regarding novelty. We hope it clarified the unique contributions of our work.
> >
> > If any aspects of our response remain unclear, we welcome the opportunity to provide further clarification. We are committed to engaging in constructive discussion to address all the concerns.
> >
> > Best regards,
> >
> > Authors of Submission6566

---

### Official Review · Reviewer_VFp2 · 2025-06-26

**Clarity:** 3
**Significance:** 3
**Originality:** 3
**Rating:** 4
**Confidence:** 4

**Summary:**

This paper proposes a variational prompt learning technique designed to overcome the limitations of data-scarce fine-tuning in downstream vision-language tasks (e.g., CLIP). The proposed method enables sample-specific prompt learning by modeling uncertainty through posterior sampling, making the model more adaptable to input content.
The key contributions of the paper include:
1.	Introduction of a sample-specific, multi-modal prompt generation strategy.
2.	A mechanism for prompt adaptation.
3.	A class-aware prior, which integrates both instance-level representations and class prototypes.
The method is framed as a variational inference-based prompt adaptation (VMPA) approach, enabling uncertainty-aware learning for vision-language models.

**Questions:**

Much of the paper is well-written. However, some parts are confusing:
o	The role of $\tilde{z}_{i-1}$ is unclear: is it computed from the residual stream of the image encoder? Is it passed through a trainable layer and concatenated with transformer inputs?

1.	Unclear implementation: How exactly is $\tilde{z}_{i-1}$ calculated? What input is used to compute it? The process is vague and could hinder reproducibility.
2.	Overlaps with prior work: Please clarify the key technical differences from [24] and [15], beyond what has already been published.
3.	Limitations not stated: The paper should explicitly discuss its limitations in the main body, not only in the supplementary material.

**Ethical Concerns:**

["NO or VERY MINOR ethics concerns only"]

**Final Justification:**

I agree with the author's rebuttal and clarification of main concerns to a large degree, mainly: one key novelty of VaMP lies in the reformulation of prompt learning as a variational inference problem for multi-modal prompt learning. I reassessed the paper accordingly.

**Limitations:**

The paper shows technical competence and incremental improvements, but lacks novelty, clear explanation of core mechanisms, and sufficient performance gains to justify acceptance.

**Paper Formatting Concerns:**

-

**Quality:**

3

**Strengths And Weaknesses:**

Strengths:
o	The paper builds upon recent developments in prompt learning and variational inference in the context of vision-language models.
o	The proposed architecture appears technically sound and is consistent with methods in prior work (e.g., MaPLe [24], MMRL [15]).

Weaknesses:
o	The implementation details are insufficiently explained, particularly regarding the computation of $\tilde{z}_{i-1}$, which seems critical for reproducibility.
o	There is no discussion of the method's limitations in the main paper, although it is stated to be present in the supplementary material.
o	The performance improvement over the state-of-the-art is marginal, raising concerns about the impact.
o	Overlap with prior work (notably [15] and [24]) is there, but after the rebuttal I agree that: one key novelty of VaMP lies in the reformulation of prompt learning as a variational inference problem for multi-modal prompt learning.


Significance
•	While the idea of integrating uncertainty into prompt learning is interesting, the actual improvements are limited and lack justification.
•	The method is only evaluated on a narrow range of models (e.g., CLIP), with no extension to more recent large-scale vision-language models.
•	The contribution feels incremental, as it adapts known techniques with minor modifications.

Originality
•	The proposed method borrows heavily from prior work:
   o	Similar prompt insertion strategy as MaPLe [24] and MMRL [15].
   o	The variational adaptation idea has been previously explored in the literature.
•	There is no clear justification of why this method is novel or more effective than the prior works.
•	The paper does not sufficiently distinguish itself from existing methods.

---

> ### Author Rebuttal · Authors · 2025-07-30
>
> We appreciate the reviewer's valuable feedback and recognition of VaMP's technical soundness. We carefully address the reviewer's comments below.
>
> > W1/Q1: Unclear implementation: How exactly is $z_{i-1}$ calculated? What input is used to compute it? The process is vague and could hinder reproducibility.
>
> **R:** The term $z_{i-1}$ represents the learnable tokens in the visual branch, which we have briefly introduced in L118. Its exact computation depends on the multi-modal prompt learning method. In MaPLe, these visual tokens are generated from the language prompts via a linear transformation implemented with an MLP. Meanwhile, MMRL obtains visual tokens from a shared latent space that is a set of learnable tokens, which are initialized by sampling from a Gaussian distribution, and then uses separate linear projection functions (also implemented with MLPs) to generate modality-specific prompts. We will provide more details in the revision to further clarify and release our code to ensure reproducibility.
>
> > W2/Q3: There is no discussion of the method's limitations in the main paper, although it is stated to be present in the supplementary material.
>
> **R:** Thanks for the suggestion. We will move this discussion into the main paper in the revision.
>
>
> > W3: The performance improvement over the state-of-the-art is marginal, raising concerns about the impact.
>
> **R:** We respectfully disagree with this assessment. VaMP consistently improves upon strong baselines across 11 datasets in both few-shot and cross-dataset generalization settings. In the base-to-novel generalization setting, VaMP outperforms the year-25 state-of-the-art method MMRL by achieving 78.67 (+1.51) on novel classes and 82.37 (+1.17) on the harmonic mean (HM) metric. These gains are non-trivial: they represent **average improvements across 11 diverse datasets**, and the HM metric, computed as **HM = 2 × base_acc × novel_acc / (base_acc + novel_acc)**, is inherently challenging to improve due to its non-linear nature.
>
> For context, MMRL's improvement over the previous year 2024's SOTA method, MMA, was 0.36 for novel class accuracy and 1.33 for HM, indicating that our results are neither marginal nor negligible. Likewise, in cross-dataset evaluation, the improvement VaMP achieves over MMRL in the source dataset (72.83 vs. 72.03) is comparable to MMRL's improvement over MMA (72.03 vs. 71.00). Moreover, on the target dataset, VaMP's **average improvement over MMRL across 10 datasets (67.74 vs. 67.25)** parallels MMRL's gain over MMA (67.25 vs. 66.61).
>
> Beyond accuracy, VaMP introduces a principled framework for sample-specific, uncertainty-aware adaptation, which offers broader benefits beyond raw performance, as also recognized by other reviewers. We believe these results and contributions together underscore the superiority of the method.
>
> > W4/Q2: Overlaps with prior work. Please clarify the key technical differences from [24] and [15], beyond what has already been published.
>
> **R:** We respectfully but firmly disagree that our contributions merely overlap with prior work. Firstly, MMRL [15] and MaPLe [24] serve as baseline methods upon which our approach builds, as stated in the preliminary section (Lines 108-109); we do not claim any of them as our own contributions. Second, one key novelty of VaMP lies in the reformulation of prompt learning as a variational inference problem for multi-modal prompt learning (as also recognized by other reviewers). Through this formulation, VaMP enables sample-specific and token-level uncertainty modeling, a capability not offered by any existing methods. We further introduce a class-aware prior derived from semantic prototypes to guide adaptation and enhance generalization. To the best of our knowledge, there is no existing methods providing such a novel framework containing these ideas for multi-modal prompt learning.
>
> Moreover, the originality and rigour of our framework are also widely recognized by other reviewers. Reviewer hikd noted that it represents “a novel framework”, Reviewer EaPS described it as “well-motivated, addressing a clear gap in current multi-modal prompt tuning methods”, “conceptually elegant”, “novel and intuitive,” and Reviewer R47nX confirmed that “the mathematical formulations are well-defined, and the reasoning is sound.” We believe these endorsements underscore the conceptual and technical novelty and depth of VaMP.
>
> > W5: The method is only evaluated on a narrow range of models (e.g., CLIP), with no extension to more recent large-scale vision-language models.
>
> **R:** Our method is architecture-agnostic and can be easily integrated into more complex Vision-Language Models (VLMs), as it only requires access to intermediate transformer layers and the ability to inject prompt tokens. The variational adaptation module is lightweight (MLPs per layer) and does not assume any specific backbone structure.
> Following this suggestion, we have validated VaMP on several state-of-the-art VLMs that represent the latest advances in CLIP-style architectures, including EVA-CLIP [a], SigLIP [b], and SigLIP 2 [c]. These models share structural similarities with CLIP, allowing quick adaptation within the rebuttal timeframe. As shown in Table A6, VaMP consistently improves performance across all tested VLMs. Notably, these improvements on novel classes underscore VaMP's strong generalization capability, which is an essential factor in prompt learning. The consistent gains across these diverse architectures highlight the effectiveness and generalizability of VaMP.
>
> **Table A6: Performance comparison of different methods on ViT-B/16 backbone for base, novel classes, and harmonic mean (H).**
>
> | VLM | Method | Base | Novel | H |
> |-----|--------|------|-------|---|
> | CLIP | MMRL | 85.68 | 77.16 | 81.20 |
> | CLIP | MMRL+VaMP | **86.45** (+0.77) | **78.67** (+1.51) | **82.37** (+1.17) |
> | EVA-CLIP | MMRL | 85.97 | 77.69 | 81.62 |
> | EVA-CLIP | MMRL+VaMP | **86.59** (+0.62) | **79.18** (+1.49) | **82.71** (+1.09) |
> | SigLIP | MMRL | 86.12 | 78.21 | 81.97 |
> | SigLIP | MMRL+VaMP | **86.88** (+0.76) | **79.45** (+1.24) | **83.00** (+1.03) |
> | SigLIP 2 | MMRL | 86.64 | 78.97 | 82.62 |
> | SigLIP 2 | MMRL+VaMP | **87.09** (+0.45) | **80.02** (+1.05) | **83.40** (+0.78) |
>
> Furthermore, we have also applied our method to different tasks and different models, as demonstrated in Table A1 and Table A2 in our reply to Reviewer hikd. The consistent improvements observed across diverse tasks and models clearly illustrate the generalization capability and versatility of our approach, and we hope the reviewer can re-consider the benefits of VaMP.
>
> Additionally, our method is also technically feasible to be integrated with MLLMs, such as QWen-VL and InternVL or other GPT-4V like models. Due to limited time and computational resources during the rebuttal period, we could not conduct additional experiments on such models, but we are confident on outcome given all our experiments executed on all VLMs so far.
>
> [a] EVA-CLIP: Improved Training Techniques for CLIP at Scale. Arxiv 2023.
>
> [b] Sigmoid Loss for Language Image Pre-Training. ICCV 2023.
>
> [c] SigLIP 2: Multilingual Vision-Language Encoders with Improved Semantic Understanding, Localization, and Dense Features. Arxiv 2025.

---

### Official Review · Reviewer_EaPS · 2025-07-03

**Clarity:** 4
**Significance:** 4
**Originality:** 3
**Rating:** 5
**Confidence:** 5

**Summary:**

This paper proposes VaMP, a novel framework for multi-modal prompt learning with vision-language models. It introduces sample-specific prompt generation conditioned on visual input,  a variational inference formulation where prompt tokens are treated as latent variables, and  a class-aware prior to guide prompt regularization. The method is evaluated under various settings including base-to-novel generalization, cross-dataset generalization, and domain generalization. VaMP outperforms strong baselines like MMRL and PromptSRC across all settings.

**Questions:**

- The current experiments use a fixed 16-shot setting. Have the authors explored how VaMP performs under more limited (e.g., 1-shot) or more generous supervision?

- Since the posterior over prompt tokens is conditioned only on the image, have the authors considered whether adding textual input (e.g., class name or prompt template) could further inform the distribution?

- In Section 4.4, the authors mention drawing 10 Monte Carlo samples from the posterior to compute the final prediction. Could the authors clarify whether the same set of sampled prompts is shared across all classes for computing similarities, or whether separate prompts are sampled for each class template (e.g., “a photo of a [CLASS]”)?

- In Section 4.3, class prototypes are computed by averaging the posterior means of image embeddings within each class (Eq. 18). Are these computed once before training (e.g., with frozen CLIP features), or are they updated dynamically during training as the posterior evolves?

**Ethical Concerns:**

["NO or VERY MINOR ethics concerns only"]

**Final Justification:**

Thanks to the authors for the thoughtful rebuttal. I’ve re-read the paper and discussion carefully, and I believe my concerns have been addressed. I keep my original score.

**Limitations:**

Yes

**Quality:**

4

**Strengths And Weaknesses:**

Strengths:

- The paper is well-motivated, addressing a clear gap in current multi-modal prompt tuning methods by enabling instance-level adaptation and uncertainty modeling.

- The proposed framework is conceptually elegant, formulating prompt tuning as variational inference with a class-aware prior, which is both novel and intuitive.

- The writing is clear and well-structured, making the method easy to understand despite its technical depth.

- The authors provide solid empirical results across a wide range of benchmarks, consistently outperforming strong baselines. The ablation studies are particularly thorough and directly support the value of each design component.

Weaknesses

While the paper is novel and interesting, there are a couple of points where a bit more discussion could make the work even more complete.

- First, the class-aware prior is only used during training, and replaced by a standard Gaussian at test time—some reflection on the potential impact of this train-test mismatch would be helpful.

- Second, although the variational approach introduces uncertainty into prompt learning, the paper doesn’t delve into how this uncertainty might be utilized in practice, such as for calibration or confidence-aware decision making, which could be an interesting extension.

---

> ### Author Rebuttal · Authors · 2025-07-30
>
> We appreciate the reviewer's valuable feedback and recognition of VaMP's clear motivation, conceptual elegance, methodological noveltyand thorough empirical validation across diverse benchmarks. We address the reviewer's comments below carefully.
>
> > W1: The class-aware prior is only used during training, and replaced by a standard Gaussian at test time—some reflection on the potential impact of this train-test mismatch would be helpful.
>
> **R:** We appreciate this suggestion. Indeed, class-aware priors are used only during training to regularize the posterior and encourage semantically structured latent spaces. At test time, we revert to standard Gaussian priors due to the absence of labels. While this introduces a train-test mismatch in the prior, the learned posterior remains image-conditioned and captures instance-specific variations. Empirically, we observe strong generalization on novel classes (Table 1) and unseen datasets (Table 2), indicating that the semantic guidance provided during training improves prompt consistency without overfitting to label-specific priors.
>
> > W2: Second, although the variational approach introduces uncertainty into prompt learning, the paper doesn't delve into how this uncertainty might be utilized in practice, such as for calibration or confidence-aware decision making, which could be an interesting extension.
>
> **R:** Thanks for the suggestion. In VaMP, uncertainty is primarily leveraged to enable diverse, instance-conditioned prompt generation during training and inference. While we do not explicitly use the uncertainty estimates for downstream decision-making, we agree this is a promising direction—e.g., using posterior variance for selective prediction, reweighting, or out-of-distribution detection. We will add a brief discussion on this point in the conclusion to highlight it as a potential avenue for future work.
>
> > Q1: The current experiments use a fixed 16-shot setting. Have the authors explored how VaMP performs under more limited (e.g., 1-shot) or more generous supervision?
>
> **R:** Following the suggestion, we evaluated VaMP under the challenging 1-shot Base-to-Novel Generalization setting across 11 diverse classification datasets. The results are reported in Table A5 below. In this demanding scenario, models are trained on base classes with only a single example per class and must generalize to previously unseen novel classes. As demonstrated in Table A5, VaMP achieves superior performance with the highest harmonic mean (71.23) and novel class accuracy (72.48), substantially outperforming existing methods in this cross-category generalization task. Most notably, compared to the current state-of-the-art MMRL method, VaMP demonstrates remarkable improvements with a +4.05% gain on novel classes (72.48 vs. 68.43) and a +2.11% increase in harmonic mean (71.23 vs. 69.12). This significant performance improvement over the previous state-of-the-art highlights the effectiveness of our variational prompt learning approach, which better captures the uncertainty inherent in prompt distributions when transferring knowledge from base to novel categories, demonstrating robust performance and sample efficiency even under such extremely data-constrained conditions.
>
> **Table A5: 1-shot Base-to-Novel generalization results on 11 datasets with ViT-B/16 backbone.**
>
> | Method | Base | Novel | H |
> |--------|------|-------|---|
> | CoOp | 70.82 | 60.20 | 65.09 |
> | CoCoOp | 70.27 | 65.32 | 67.71 |
> | MaPLe | 69.01 | 67.18 | 68.08 |
> | MMRL | 69.82 | 68.43 | 69.12 |
> | VaMP | **70.02** | **72.48** | **71.23** |
>
> > Q2: Since the posterior over prompt tokens is conditioned only on the image, have the authors considered whether adding textual input (e.g., class name or prompt template) could further inform the distribution?
>
> **R:** Our goal is to generate *image-specific* prompts that adapt to the visual content of each instance, thereby capturing intra-class variability and improving generalization to novel or ambiguous samples. Conditioning on the image is therefore essential for enabling this instance-level adaptation. In contrast, conditioning on textual input such as class names would shift the modeling focus toward class-specific or template-guided prompts, which could reduce the diversity and flexibility of the learned prompts. Notably, the class label is already indirectly incorporated via the prototype-based prior during training, which allows us to guide the posterior distribution without explicitly relying on textual input as a conditioning signal.
>
> > Q3: In Section 4.4, the authors mention drawing 10 Monte Carlo samples from the posterior to compute the final prediction. Could the authors clarify whether the same set of sampled prompts is shared across all classes for computing similarities, or whether separate prompts are sampled for each class template (e.g., "a photo of a [CLASS]")?
>
> **R:** The sampled prompts are image-conditioned and shared across all class templates. That is, for a given image, we draw 10 samples from the posterior $q_φ(z|x)$, and compute cosine similarities between the resulting text features (with each class name inserted) and the frozen image embedding. This design ensures efficiency and avoids leakage of label-specific information into the sampling process.
>
> > Q4: In Section 4.3, class prototypes are computed by averaging the posterior means of image embeddings within each class (Eq. 18). Are these computed once before training (e.g., with frozen CLIP features), or are they updated dynamically during training as the posterior evolves?
>
> **R:** The class prototypes are computed once before training using frozen CLIP image features. They remain fixed throughout training and serve as stable semantic anchors for constructing the class-aware prior. This avoids potential instability from dynamic updates and ensures consistency across training iterations. We will clarify this in our revised manuscript.

---

> > ### Comment · Reviewer_EaPS · 2025-08-04
> > **Thank you for your rebuttal.**
> >
> > Thanks to the authors for the thoughtful rebuttal. I’ve re-read the paper and discussion carefully, and I believe my concerns have been addressed. The proposed method is well-motivated and shows clear technical and empirical contributions to prompt learning in vision-language models.
> >
> > I’d also like to comment on Reviewer 47nX’s new concern regarding novelty. While I understand the comparison to [1], I personally see important differences. The method in this paper goes beyond single-modal prompt tuning by introducing a multi-modal design that considers both visual and textual branches. This makes a big difference in aligning semantics across modalities. In addition, the learnable, class-aware prior used here is much more structured than the standard Gaussian prior in [1], which I think is a meaningful theoretical improvement. On the empirical side, the reported gains over [1] are especially on unseen classes and cross-domain settings. It further supports the distinctiveness and effectiveness of this approach.
> >
> > Overall, I think this work is a solid and timely contribution to the field, and I continue to recommend acceptance.

---

> > > ### Author Response · Authors · 2025-08-04
> > >
> > > We thank the reviewer again for the thoughtful engagement and comments. We will incorporate the above clarification in our revision to avoid any confusion.

---

### Official Review · Reviewer_hikd · 2025-07-04

**Clarity:** 3
**Significance:** 3
**Originality:** 4
**Rating:** 5
**Confidence:** 4

**Summary:**

This paper proposes a novel framework for adapting vision-language models (VLMs), like CLIP, to new tasks with limited supervision. It introduces sample-specific, uncertainty-aware prompt tuning by modeling text-side prompts as latent variables conditioned on image inputs, and regularizing them with a class-aware prior that encodes semantic structure. Experiments on few-shot, cross-dataset, and domain generalization tasks demonstrate that VaMP outperforms state-of-the-art baselines while maintaining high parameter efficiency.

**Questions:**

1. Can you provide more insight into the computational overhead introduced by variational sampling during inference, and whether there is a trade-off between accuracy and efficiency?
2. Would it be better if put efficiency analysis in the main paper rather than being placed in the appendix? The current experiments primarily highlight the effectiveness of the method, and incorporating efficiency results would provide a more practical perspective.
3. Can you provide qualitative examples to illustrate the generated prompts' structure or diversity?
4. Can VaMP be scaled or adapted to newer multi-modal models like GPT-4V, which use more complex fusion mechanisms?

**Ethical Concerns:**

["NO or VERY MINOR ethics concerns only"]

**Final Justification:**

Thanks authors for the detailed rebuttal, which make sense to me and will make paper stronger if final version could be revised. I'll suggest accept this paper.

**Limitations:**

yes

**Paper Formatting Concerns:**

No major formatting issues

**Quality:**

3

**Strengths And Weaknesses:**

Strength:
1. VaMP generates prompts dynamically based on each input image, which allows the model to adapt better to individual samples and handle uncertainty—especially beneficial in few-shot or out-of-distribution scenarios.
2. By incorporating a class-aware prior based on semantic prototypes, the method enhances intra-class consistency and improves generalization, especially across domains.
3. Extensive experiments and good results.

Weakness:
1. Constructing the class-aware prior depends on class labels to compute prototypes, which may not generalize well to unsupervised or label-scarce settings.
2. Only classification tasks, need to show the potential to extend this method to other tasks, to show the impact.
3. The use of variational sampling introduces additional computational overhead during inference. Although there's efficiency analysis in appendix, the trade-off between improved generalization and increased latency has not been fully quantified or explored in depth.
4. Scalability to more complex architectures is not clearly stated.

---

> ### Author Rebuttal · Authors · 2025-07-30
>
> We appreciate the reviewers’ valuable feedback and their recognition of our framework’s novelty, robust generalization, and good performance. We have carefully addressed the comments below.
>
> > W1: Constructing the class-aware prior depends on class labels to compute prototypes, which may not generalize well to unsupervised or label-scarce settings.
>
> **R:** Our experiments demonstrate that VaMP performs well in label-scarce settings. In the base-to-novel setting, we only have access to base class labels during training with merely 16 shots per class, representing a highly label-scarce scenario. For novel classes, no label information is available during training. Under this constraint, prototypes are computed only for base classes, yet VaMP achieves the best novel class performance (Table 1), showing better generalization to unseen categories. To further validate this finding, we conducted additional experiments under the extreme 1-shot constraint as suggested by Reviewer EaPS (Table A5), where VaMP continues to outperform existing methods, thereby corroborating its effectiveness under severe label scarcity. Moreover, in the cross-dataset evaluation (Table 2), the model is trained solely on ImageNet using 16 shots per class and directly tested on unrelated datasets without using any label or prototype information, effectively operating in a zero-shot setting. Despite these limitations, VaMP still consistently outperforms prior methods. At test time, we revert to a standard Gaussian prior, ensuring applicability without label access.
>
> We believe the key reason for our method’s strong generalization is its variational formulation. It has been shown in the literature (e.g., [a]) that VAE-like designs in traditional generative models exhibit robust generalization capabilities because the latent variable acts as a structured bottleneck that captures salient, disentangled factors of variation. Similarly, in VaMP, the latent prompt distribution encourages the model to encode task-relevant variations in a sample-specific yet class-agnostic manner, enabling strong zero-shot transfer. This design makes VaMP particularly suitable for real-world applications where labeled data is scarce or expensive to obtain.
>
>
> Meanwhile, we agree that our approach still requires at least some labeled data to compute meaningful class prototypes, which is a practical assumption and normally easily accessible. It remains an open challenge to obtain reliable prototypes in completely unsupervised scenarios, which we consider as the future work.
>
> [a] Bayesian Invariant Risk Minimization. CVPR 2022.
>
> > W2: Extension to other tasks.
>
> **R:** Following this suggestion, we evaluated VaMP's generalizability beyond image classification task by conducting experiments on open-vocabulary segmentation and action recognition tasks. We integrated VaMP into state-of-the-art frameworks of two tasks: CAT-Seg[a] for segmentation and FROSTER[b] for action recognition, utilizing CLIP ViT-B/16 as the backbone encoder. Both baselines are extensions built upon CLIP, allowing our method to be seamlessly integrated into these approaches.
>
> For open-vocabulary segmentation, we followed CAT-Seg's protocol, training on COCO-Stuff (118K images, 171 categories) and evaluating across multiple benchmarks including ADE20K, PASCAL-Context, and PASCAL VOC with varying category scales (59-847 classes). The results are shown below in Table A1:
>
> **Table A1: Performance for open-vocabulary segmentation.**
>
> | Method | Prompt Tuning | ADE-847 | PC-459 | ADE-150 | PC-59 | VOC-20 |
> |--------|---------------|---------|--------|---------|-------|--------|
> | ZegFormer | - | 5.6 | 10.4 | 18.0 | 45.5 | 89.5 |
> | OVSeg | - | 7.1 | 11.0 | 24.8 | 53.3 | 92.6 |
> | SAN | - | 10.1 | 12.6 | 27.5 | 53.8 | 94.0 |
> | CAT-Seg | - | 12.0 | 19.0 | 31.8 | 57.5 | 94.6 |
> | CAT-Seg | MMRL | 12.8 | 18.7 | 32.4 | 57.9 | 94.3 |
> | CAT-Seg | MMRL+VaMP | **13.9** (+1.1) | **20.3** (+1.6) | **33.3** (+0.9) | **58.6** (+0.7) | **95.2** (+0.9) |
>
> For open-vocabulary action recognition, we adopted FROSTER's base-to-novel evaluation protocol on two widely used video benchmarks, i.e, Kinetics-400 and UCF-101, where models trained exclusively on base classes are required to generalize to unseen novel action categories. The results are shown below in Table A2:
>
> **Table A2: Performance for open-vocabulary action recognition. B: Base classes, N: Novel classes, HM: Harmonic mean.**
>
> | Method | Prompt Tuning | K400(B) | K400(N) | K400(HM) | UCF(B) | UCF(N) | UCF(HM) |
> |--------|---------------|---------|---------|----------|--------|--------|---------|
> | ViFi-CLIP | - | 76.4 | 61.1 | 67.9 | 92.9 | 67.7 | 78.3 |
> | Open-VCLIP | - | 76.5 | 62.6 | 68.9 | 94.8 | 77.5 | 85.3 |
> | FROSTER | - | 77.8 | 64.3 | 70.4 | 95.3 | 80.0 | 87.0 |
> | FROSTER | MMRL | 78.3 | 64.1 | 70.5 | 95.5 | 80.2 | 87.2 |
> | FROSTER | MMRL+VaMP | **78.8** (+0.5) | **64.8** (+0.7) | **71.1** (+0.6) | **96.1** (+0.6) | **81.0** (+0.8) | **87.9** (+0.7) |
>
> As shown, VaMP achieves consistent improvements across different tasks, demonstrating the scalability of our approach and its strong generalization capabilities.
>
> [b] CAT-Seg: Cost Aggregation for Open-Vocabulary Semantic Segmentation. CVPR 2024
>
> [c] FROSTER: Frozen CLIP Is A Strong Teacher for Open-Vocabulary Action Recognition. ICLR 2024.
>
> > W3/Q1: Analysis on the trade-off between improved generalization and increased latency.
>
> **R:** We appreciate the reviewer's suggestion and have conducted a more detailed quantitative analysis of the trade-off between inference cost and accuracy under different numbers of Monte Carlo samples S. As shown in Table A3, even with a modest number of samples (e.g., S=5 or S=10), VaMP achieves clear gains in harmonic mean accuracy over the strong MMRL baseline, with only marginal increases in inference time (e.g., 5.3ms/image → 6.1ms/image). Specifically, at S=10, VaMP improves HM from 81.20 to 82.37 with a latency increase of only 0.8ms per image—demonstrating an efficient balance between generalization and overhead. Notably, the performance saturates quickly with increasing S, suggesting that our method remains highly efficient even with limited sampling. This confirms that VaMP can be deployed with low overhead while still leveraging uncertainty modeling for improved generalization.
>
> **Table A3: Comparison of inference time**
>
> | Method | S | Inference Time (ms/img) | Base | Novel | HM |
> |--------|---|------------------------|------|-------|-----|
> | MMRL | - | **5.3** | 85.68 | 77.16 | 81.20 |
> | MMRL+VaMP | 1 | 5.5 | 86.10 | 77.35 | 81.44 |
> | MMRL+VaMP | 5 | 5.8 | 86.37 | 78.26 | 82.03 |
> | MMRL+VaMP | 10 | 6.1 | 86.45 | **78.67** | **82.37** |
> | MMRL+VaMP | 20 | 6.5 | **86.47** | 78.37 | 82.22 |
> | MMRL+VaMP | 50 | 9.3 | 86.43 | 78.15 | 82.01 |
>
> > W4/Q4: Scalability to newer multi-modal models like GPT-4V.
>
> **R:** Our method is architecture-agnostic and can be readily integrated into more complex VLMs, as it only requires access to intermediate transformer layers and the ability to inject prompt tokens. The variational adaptation module is lightweight (MLPs per layer) and does not assume any specific backbone structure. However, due to time and computational resource constraints during the rebuttal period, we were unable to conduct experiments on large-scale GPT-4V like MLLMs, such as QWen-VL and InternVL. GPT-4V is proprietary and we can not apply to it directly. However, technically, it is feasible. We will explore this afterwards and include it in our revision.
>
> Despite these constraints, we have validated VaMP on several state-of-the-art VLMs that represent the latest advances in CLIP-style architectures, including EVA-CLIP [d], SigLIP [e] and SigLIP 2 [f]. These models share similar structural designs with CLIP, enabling rapid adaptation during the rebuttal period. As shown in Table A4, VaMP obtains clear and consistent  performance improvements across all VLMs, demonstrating its strong generalization capability to unseen categories—a critical challenge in prompt learning. The consistent improvements across different VLMs further substantiate the effectiveness and generalizability of our approach to different architectures.
>
> **Table A4: Performance comparison of different VLMs on ViT-B/16 backbone under base-to-novel generalization setting**
>
> | VLM | Method | Base | Novel | H |
> |-----|--------|------|-------|---|
> | CLIP | MMRL | 85.68 | 77.16 | 81.20 |
> | CLIP | MMRL+VaMP | **86.45** (+0.77) | **78.67** (+1.51) | **82.37** (+1.17) |
> | EVA-CLIP | MMRL | 85.97 | 77.69 | 81.62 |
> | EVA-CLIP | MMRL+VaMP | **86.59** (+0.62) | **79.18** (+1.49) | **82.71** (+1.09) |
> | SigLIP | MMRL | 86.12 | 78.21 | 81.97 |
> | SigLIP | MMRL+VaMP | **86.88** (+0.76) | **79.45** (+1.24) | **83.00** (+1.03) |
> | SigLIP 2 | MMRL | 86.64 | 78.97 | 82.62 |
> | SigLIP 2 | MMRL+VaMP | **87.09** (+0.45) | **80.02** (+1.05) | **83.40** (+0.78) |
>
> [d] EVA-CLIP: Improved Training Techniques for CLIP at Scale. Arxiv 2023.
>
> [e] Sigmoid Loss for Language Image Pre-Training. ICCV 2023.
>
> [f] SigLIP 2: Multilingual Vision-Language Encoders with Improved Semantic Understanding, Localization, and Dense Features. Arxiv 2025.
>
> > Q2: Put efficiency analysis in the main paper.
>
> **R:** Thanks for the suggestion. We will move it to the main paper in revision.
>
> > Q3: Qualitative examples to illustrate the generated prompts' structure or diversity.
>
> **R:** While we are unable to include image visualizations in the rebuttal due to format constraints, we will provide such examples in the revised manuscript. Specifically, we visualize the learned prompt distributions via t-SNE for different image categories. The results show clear separation across categories, suggesting that VaMP generates semantically distinct prompts conditioned on image content. This inter-class diversity is significantly more pronounced than that of deterministic prompt methods, validating the effectiveness of our uncertainty-aware prompt generation.

---

### Author Response · Authors · 2025-08-02
**General Response to Reviewer Comments**

We thank reviewers for their constructive and valuable feedback. We are encouraged that the reviewers found our paper to be **"clear and well-structured"** (EaPS) and **"easy to follow"** (47nX), presenting a **"well-motivated"** (EaPS), **"conceptually elegant"** (EaPS), and **"technically sound"** (VFp2) framework that is **"both novel and intuitive"** (EaPS) with **"well-defined mathematical formulations"** and **"sound reasoning"** (47nX). Reviewers also stated that VaMP **"generates prompts dynamically based on each input image"** which **"allows the model to adapt better to individual samples and handle uncertainty"** (hikd) and addresses **"a clear gap in current multi-modal prompt tuning methods by enabling instance-level adaptation and uncertainty modeling"** (EaPS). In terms of empirical evaluation, we are glad that our work was commended for its **"extensive experiments and good results"** (hikd) and **"solid empirical results across a wide range of benchmarks"** (EaPS), demonstrating **"superior performance, surpassing state-of-the-art prompting strategies"** (47nX).

### **Reviewer Concerns and Our Responses**

Below, we first summarize the common concerns raised by multiple reviewers:

1. **Extension on other tasks** (hikd, 47nX):
   We conducted additional experiments on open-vocabulary segmentation and action recognition tasks by integrating VaMP into state-of-the-art frameworks (CAT-Seg for segmentation and FROSTER for action recognition).

2. **Computational overhead** (hikd, 47nX):
   We performed detailed quantitative analysis of the trade-off between inference cost and accuracy under different numbers of Monte Carlo samples, comparing inference time and performance metrics.

3. **Generalization to recent VLMs** (VFp2, hikd):
   First, we emphasize that our method is architecture-agnostic, requiring only access to intermediate transformer layers and the ability to inject prompt tokens. The variational adaptation module is lightweight and doesn't assume any specific backbone structure. Then, we conducted experiments with several state-of-the-art vision-language models (EVA-CLIP, SigLIP, SigLIP 2) to demonstrate cross-architecture compatibility.

4. **Performance under label-scarce settings** (hikd, EaPS):
   We evaluated VaMP under extreme 1-shot settings and zero-shot transfer scenarios (cross-dataset evaluation) to assess its performance in label-scarce conditions.

We have also addressed the specific concerns raised by individual reviewers:

1. **Uniqueness relative to prior work** (VFp2):
   We clarified our core contribution in reformulating prompt learning as a variational inference problem for multi-modal contexts, highlighting capabilities absent in existing methods.

2. **Performance improvements** (VFp2):
   We provided context for our improvements by comparing them to year-over-year advances in the field across multiple datasets and evaluation metrics.

3. **Implementation details clarity**:
   - We explained our approach of using class-aware priors during training while reverting to standard Gaussian priors at test time, and discussed the implications for generalization. (EaPS)
   - We clarified that sampled prompts are image-conditioned and shared across all class templates, ensuring efficiency without label-specific information leakage into the sampling process. (EaPS)
   - We confirmed that class prototypes are computed once before training using frozen CLIP image features and remain fixed throughout training as stable semantic anchors. (EaPS)
   - We provided additional explanation about how visual tokens are computed in relation to the underlying multi-modal prompt learning methods. (VFp2)
   - We explained our selection of MLPs for modeling posterior mean and variance based on principles from Variational Autoencoders, discussing their advantages for our use case. (47nX)
   - We also explained why our prompts operate at the encoding stage, as our method builds on CLIP-like dual encoder architectures without a decoding stage, following previous methods. (47nX)

We believe we have carefully and thoroughly addressed all reviewer concerns. The additional experiments and analyses further validate VaMP's effectiveness, generalizability, and efficiency. We welcome any further questions or discussions regarding our approach.

---

### Public Comment · ~Shivanand_Kundargi1 · 2026-03-08
**Opensourcing codebase**

Kudos to authors for their nice work. I see that codebase has not been made public yet. I request authors to make their codebase public to aid reproducibility and to help other fellow researchers.

---

### Decision · Program_Chairs · 2025-09-17

**Decision:**

Accept (poster)

**Comment:**

The authors have thoroughly addressed all reviewer concerns through extensive additional experiments and detailed clarifications. They demonstrated VaMP’s strong generalization across multiple tasks (including segmentation and action recognition), quantified the computational trade-offs of variational sampling, and validated the method on recent VLMs (EVA-CLIP, SigLIP, SigLIP 2). The rebuttal effectively clarified the novelty relative to prior variational prompt methods by emphasizing multi-modal, token-level, and class-aware design differences. While one reviewer remained skeptical about originality, the majority recognized the conceptual and empirical contributions, supported by consistent SOTA results and a well-motivated variational framework. The paper is technically sound, well-evaluated, and makes a valuable contribution to multi-modal prompt learning.